# A survey and benchmark of high-dimensional Bayesian optimization of discrete sequences

**Miguel González-Duque**[*]
University of Copenhagen

**Richard Michael**
University of Copenhagen

**Simon Bartels**
Paul Sabatier University Toulouse

**Yevgen Zainchkovskyy**
Technical University of Denmark

**Søren Hauberg**[*]
Technical University of Denmark

**Wouter Boomsma**[*]
University of Copenhagen

## Abstract

Optimizing discrete black box functions is key in several domains, e.g. protein engineering and drug design. Due to the lack of gradient information and the need for sample efficiency, Bayesian optimization is an ideal candidate for these tasks. Several methods for high-dimensional continuous and categorical Bayesian optimization have been proposed recently. However, our survey of the field reveals highly heterogeneous experimental set-ups across methods and technical barriers for the replicability and application of published algorithms to real-world tasks. To address these issues, we develop a unified framework to test a vast array of high-dimensional Bayesian optimization methods and a collection of standardized black box functions representing real-world application domains in chemistry and biology. These two components of the benchmark are each supported by flexible, scalable, and easily extendable software libraries (`poli` and `poli-baselines`), allowing practitioners to readily incorporate new optimization objectives or discrete optimizers. Project website: https://machinelearninglifescience.github.io/hdbo_benchmark.

## 1 Introduction

Optimizing an unknown and expensive-to-evaluate function is a frequent problem across disciplines (Shahriari et al., 2016), examples are finding the right parameters for machine learning models or simulators, drug discovery (Gómez-Bombarelli et al., 2018; Griffiths and Hernández-Lobato, 2020; Pyzer-Knapp, 2018), protein design (Stanton et al., 2022; Gruver et al., 2023), hyperparameter tuning in Machine Learning (Snoek et al., 2012; Turner et al., 2021) and train scheduling. In some scenarios, evaluating the black box involves an expensive process (e.g. training a large model, or running a physical simulation); Bayesian Optimization (BO, Močkus (1975)) is a powerful method for sample efficient black box optimization. High dimensional (discrete) problems have long been identified as a key challenge for Bayesian optimization algorithms (Wang et al., 2013; Snoek et al., 2012) given that they tend to scale poorly with both dataset size and dimensionality of the input.

High-dimensional BO has been the focus of an entire research field (see Fig. 1), in which methods are extended to address the curse of dimensionality and its consequences (Binois and Wycoff, 2022; Santoni et al., 2023). Within this setting, discrete sequence optimization has received particular focus,

---

[*]emails: miguelgondu@gmail.com, sohau@dtu.dk, wb@di.ku.dk

38th Conference on Neural Information Processing Systems (NeurIPS 2024) Track on Datasets and Benchmarks.

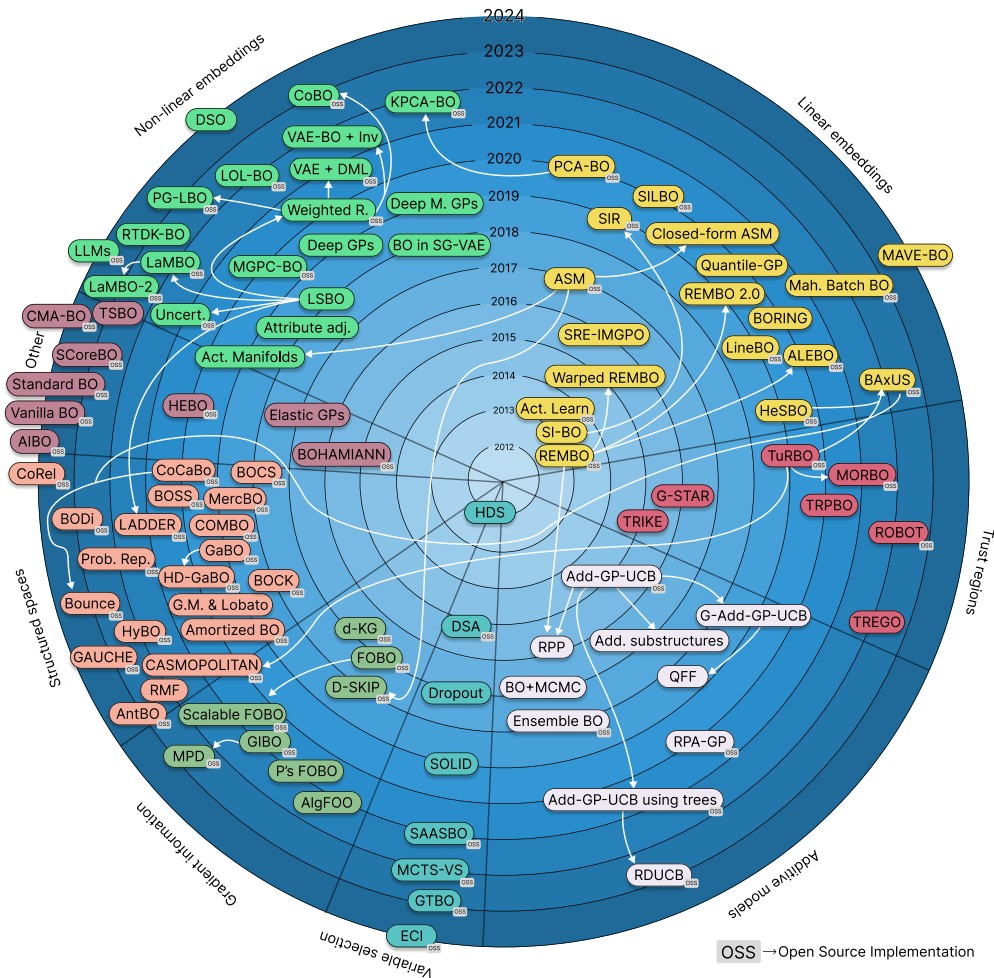

Figure 1: A timeline of high-dimensional Bayesian optimization methods, with arrows drawn between methods that explicitly augment or use each other. References can be found in supplementary Table 4. The figure is inspired by Justesen et al. (2020). An interactive version can be found in our project page.

due to its applicability in the optimization of molecules and proteins. However, prior work often focuses on sequence lengths and number of categories below the hundreds (see Fig. 2), making it difficult for practitioners to judge expected performance on real-world problems in these domains. We contribute (i) a survey of the field while focusing on the real-world applications of high-dimensional discrete sequences, (ii) a benchmark of several optimizers on established black boxes, and (iii) an open source, unified interface: `poli` and `poli-baselines`.[2]

## 2   Preliminaries

**Bayesian Optimization and Gaussian processes.**   Bayesian optimization requires a surrogate model and an acquisition function (Garnett, 2023). Given both, the objective function is sequentially optimized by fitting a model to the given observations and numerically optimizing the acquisition function with respect to the model to select the next configuration for evaluation. Frequently, the model is a Gaussian process (GP, Rasmussen and Williams (2006)), and popular choices for the acquisition function are *Expected Improvement* (Jones et al., 1998; Garnett, 2023) and the *Upper Confidence Bound* (Srinivas et al., 2012). A GP allows to express a prior belief over functions.

---

[2]`https://github.com/MachineLearningLifeScience/{`poli`, `poli-baselines`}`

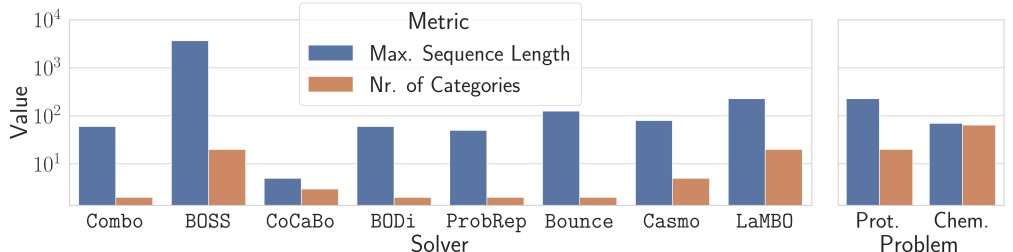

Figure 2: Existing BO methods tackle problems with insufficiently low effective dimensions. This figure shows sequence length and nr. of categories of the highest search space in the original tests. For reference, the discrete optimization problems usually tackled by practitioners in chemistry and biology are of the order of $10^2$ in sequence length, and $> 10^1$ in nr. of categories. Methods that optimize directly in discrete space (e.g. BODi, ProbRep, Bounce; Sec. 3.7) are tested in lower sequence lengths and dictionary sizes; methods that rely on unsupervised information (e.g. LaMBO, etc.; Sec. 3.5) are able to optimize more complex problems, like protein engineering or small molecule optimization.

Formally, it is a collection of random variables, such that every finite subset follows a multivariate normal distribution, described by a mean function $\mu$, and a positive definite covariance function (kernel) (Rasmussen and Williams, 2006, p. 13). Assuming that observations of the function are distorted by Gaussian noise, the posterior over the function conditioned on these observations is again Gaussian. The prediction equations have a closed form and can be evaluated in $\mathcal{O}(N^3)$ time where $N$ is the number of observations.

**Is high-dimensional Bayesian Optimization difficult?** There are three reasons why BO is thought to scale poorly with dimension. The first reason is that GPs fail to properly fit to the underlying objective function in high dimensions. Secondly, even if the GPs were to fit well there is still the problem of optimizing the high-dimensional acquisition function. Finally, Gaussian Processes are believed to scale poorly with the size of the dataset, limiting us to low-budget scenarios (Binois and Wycoff, 2022). Folk knowledge suggests that GPs fail to fit functions above the meager limit of $\sim 10^1$ dimensions (Santoni et al., 2023) and $\sim 10^4$ datapoints.

Hvarfner et al. (2024) recently disputed these well-entrenched narratives by showing that poor GP fitting could be caused by a poor choice of regularizer; mitigating the curse of dimensionality could be as easy as including a dimensionality-dependent prior over lengthscales. Furthermore, Xu and Zhe (2024) argues that even the simplest BO outperforms highly elaborate methods.

**Optimization of discrete sequences & applications.** Most HDBO methods are tested on toy examples, hyperparameter tuning, or reinforcement learning tasks (Binois and Wycoff, 2022; Penubothula et al., 2021). We focus on discrete sequence optimization, which has several applications beyond the usual examples (e.g. MaxSAT, or PestControl) (Papenmeier et al., 2024), and is key in applications to biology and bioinformatics (Gómez-Bombarelli et al., 2018; Stanton et al., 2022; Gruver et al., 2023). Drug design and protein engineering can be thought of as sequence optimization problems, if we consider the SMILES/SELFIES representation of small molecules (Weininger, 1988; Krenn et al., 2020), or the amino acid sequence representation of proteins (Needleman and Wunsch, 1970).

**Related work.** Binois and Wycoff (2022) initially surveyed the field of high-dimensional GPs, focusing on applications to BO, and proposed a taxonomy of structural assumptions for GPs that includes *variable selection*, *additive models*, *linear*, and *non-linear embeddings*. This work has since been updated by Wang et al. (2023) and Santoni et al. (2023). The latter presents an empirical study on the continuous, toy-problem setting up to 60 dimensions and refines the taxonomy (Binois and Wycoff, 2022) into five categories, separating *trust regions* from the rest. Griffiths et al. (2023) compare different kernel functions used with binary vector representations of molecules and for the same application Kristiadi et al. (2024) study the use of different large language models. Our work is most similar to Dreczkowski et al. (2023)'s comprehensive overview of discrete BO (MCBO), and Gao et al. (2022)'s benchmark of small molecule optimization. In both, HDBO is not in focus.

# 3 A taxonomy of high-dimensional Bayesian Optimization

We describe the field of high dimensional BO and the large number of related publications through a refined taxonomy building on previous work, discussing *variable selection*, *additive models*, *trust regions*, *linear embeddings*, *non-linear embeddings*, *gradient information*, *structured spaces*, and others in turn. While encompassing taxonomies over fields may initially appear ill-advised (Wilkins, 1668, pp.22), we highlight commonalities in strategies that give structure to the HDBO problem-space.

We expand previous surveys (Binois and Wycoff, 2022; Santoni et al., 2023) and identify a finer taxonomy of seven method groups and new families of *structured spaces* (i.e. methods that work directly on mixed representations, or Riemannian manifolds, previously categorized as *non-linear embeddings*), and methods that rely on predicted *gradient information*. This new separation emphasizes the heterogeneous nature of discrete solvers: some optimizers work directly on discrete space (*structured spaces*), while others optimize using latent representations (*non-linear embeddings*); gradient-based methods are separated to show alternatives when first-order information is available or modelable. Fig. 1 presents a timeline of HDBO methods, split into these families, and all methods are detailed in supplementary Table 4; methods are grouped according to their most dominant feature.

## 3.1 Variable selection

To solve a high-dimensional problem, one approach is to focus on a subset of variables of high interest.[3] One selects the variables either by using domain expertise, or by Automatic Relevance Detection (ARD) (Rasmussen and Williams, 2006, pp.106-107) i.e. large lengthscales indicate independence under the covariance matrix for GPs. Examples of this approach include Hierarchical Diagonal Sampling (HDS) (Chen et al., 2012) and the Dimension Scheduling Algorithm (DSA) (Ulmasov et al., 2016). The former determines the active variables by a binary tree of subsets of $\{1, \ldots, D\}$, and fits GPs in lower-dimensional projections. DSA constructs a probability distribution by the principal directions of the training inputs $\{(\boldsymbol{x}_n, y_n)\}_{n=1}^N$ and subsamples the dimensions accordingly. In contrast Li et al. (2018) randomly sample subsets of active dimensions.

Other methods rely on placing priors on their lengthscales, followed by a Bayesian treatment of the training. In Sequential Optimization of Locally Important Directions (SOLID), lengthscales are weighted by a Bernoulli distributed parameter, and coordinates are removed when their posterior probability goes below a user-specified threshold. (Winkel et al., 2021). Eriksson and Jankowiak (2021) consider the Sparse Axis-Aligned Subspace (SAAS) model of a GP, restricting the function space through a (long-tailed) half-Cauchy prior on the inverse-lengthscales of the kernel.

## 3.2 Additive models

Additive models assume that the objective function $f$ can be decomposed into a sum of lower-dimensional functions. Symbolically, the coordinates of a given input $\boldsymbol{x} = (x_1, \ldots, x_D)$ are split into $M$ usually disjoint subgroups $g_1, \ldots g_M$ of smaller size, called a decomposition. Instead of fitting a GP to $D$ variables in $f$, the algorithm fits $M$ GPs to the restrictions $f|_{g_1}, \ldots f|_{g_M}$ and adds their Upper Confidence Bound. The differences between the algorithms in this family are on how the subgroups are constructed, how the additive structure is approximated, the training of the Gaussian Process, or leveraging special features (Mutny and Krause, 2018).

Han et al. (2021) select the decomposition which maximizes the marginal likelihood from a collection of randomly sampled decompositions, updating it every certain number of iterations. Alternatives include: leveraging a generalization based on restricted projections (Li et al., 2016), discovering the additive structure using model selection and Markov Chain Monte Carlo (Gardner et al., 2017), considering overlapping groups (Rolland et al., 2018), ensembles of Mondrian space-tiling trees (Wang et al., 2018), or use random tree-based decompositions (Ziomek and Bou Ammar, 2023).

## 3.3 Trust regions

Some BO algorithms restrict the evaluation of the acquisition function to a small region of input space called a *trust region*, which is centered at the incumbent and is dynamically contracted or expanded according to performance (Regis, 2016; Pedrielli and Ng, 2016; Eriksson et al., 2019).

---

[3]Under the assumption that there exists an axis-aligned lower-dimensional *active subspace*.

Contemporary variants extend to the multivariate setting (e.g. MORBO (Daulton et al., 2022a)), to quality-diversity (Maus et al., 2023) and to the optimization of mixed variables (CASMOPOLITAN by Wan et al. (2021)), including categorical. Since the trust region framework involves only the optimization of the acquisition function, several other methods leverage it alongside other structural assumptions like linear/non-linear embeddings (e.g. Tripp et al. (2020); Papenmeier et al. (2022)).

## 3.4 Linear embeddings

Instead of optimizing directly in input space $\mathbb{R}^D$, several methods rely on optimizing in a lower-dimensional space $\mathbb{R}^d$, which is linearly embedded into data space using a linear transformation $A \in \mathbb{R}^{D \times d}$ (Wang et al., 2016). The matrix $A$ can be either selected at random (Wang et al., 2016; Qian et al., 2016), computed as a low-rank approximation of the input data matrix (Djolonga et al., 2013; Zhang et al., 2019; Raponi et al., 2020), constructed using gradient information and active subspaces (Palar and Shimoyama, 2017; Wycoff et al., 2021), or through the minimization of variance estimates (Hu et al., 2024).

These methods are limited by how low-dimensional exploration translates into high dimensions. One choice of embedding matrix $A$ spans a *fixed*, highly-restricted subspace of $\mathbb{R}^D$. For this approach several issues regarding back-projections need to be addressed. Indeed, projecting from bounded domains $\mathcal{Z} \subseteq \mathbb{R}^d$ to $\mathbb{R}^D$ might render points outside the bounded domain in the input (Binois and Wycoff, 2022). Finally, the transformation $A$ is not injective, meaning a point in input space can correspond to several latent points (Binois et al., 2015; Moriconi et al., 2020b).

Binois et al. (2015) propose a kernel that alleviates these issues by including a back-projection to the bounded domain that respects distances in the embedded space. *Hashing matrices $S \in \mathbb{R}^{D \times d}$* are an alternative way to reconstruct an input data point in a bounded domain $\boldsymbol{x} \in [-1, 1]^D \subseteq \mathbb{R}^D$ from a latent point $\boldsymbol{z} \in \mathbb{R}^D$, whose entries are either 0, 1, and -1. Thus, the result of multiplying $S\boldsymbol{z}$ is a linear combination of the coordinates of $\boldsymbol{z}$ where the coefficients are 1 and -1 (Nayebi et al., 2019). These ideas have been combined with trust regions both in the continuous (Papenmeier et al., 2022) and mixed-variable settings (Papenmeier et al., 2024). A natural extension considers a family of nested subspaces, progressively growing the embedding matrix until it matches the input dimensionality (Papenmeier et al., 2022). An alternative that does not deal with reconstruction mappings (thus circumventing the aforementioned issues) uses the information learned in the lower dimensional space to perform optimization directly in input space (Horiguchi et al., 2022).

## 3.5 Non-linear embeddings

Several methods have considered non-linear embeddings to incorporate learned latent representations. One set of examples are deep latent variable models like Generative Adversarial Networks (Goodfellow et al., 2014), or variants of Autoencoders (Kingma and Welling, 2014; Stanton et al., 2022; Maus et al., 2022), algorithms that allow for modelling arbitrarily structured inputs. This is highly relevant for optimizing sequences, which are modeled as samples from a categorical distribution.

Gómez-Bombarelli et al. (2018) pioneered latent space optimization (LSBO) by learning a latent space of small molecules through their SMILES representation using a Variational Autoencoder (VAE, Kingma and Welling (2014); Rezende et al. (2014)), and optimizing metrics such as the qualitative estimate of druglikeness (QED) therein. Several approaches have followed, including usage of *a-priori* given labelled data (Eissman et al., 2018) or decoder uncertainty (Notin et al., 2021), smart retraining schemes that focus on promising points (Tripp et al., 2020), metric-learning approaches that match promising points together (Grosnit et al., 2021), constraining the latent space (Griffiths and Hernández-Lobato, 2020), latent spaces mapping to graphs (Kusner et al., 2017; Jin et al., 2018) and jointly learning the surrogate model and the latent representation (Maus et al., 2022; Lee et al., 2023; Chen et al., 2024; Kong et al., 2024). Stanton et al. (2022) take this further by learning multiple representations: one shared and required for both the decoder and surrogate, and one discriminative encoding as input for a GP used in the acquisition function. A prerequisite for these methods is a large dataset of *unsupervised* inputs, which may not be available in all applications. The methods that rely on training both the representation and the regression at the same time need *supervised* labels, which may be potentially unavailable. Optimization in embedding spaces greatly increases the complexity of problems that can be tackled, making it an appealing alternative for discrete sequence optimization in real-world tasks (see Fig. 2).

## 3.6 Gradient information

High-dimensional problems can become significantly easier when derivative information is available. Even when the objective's derivatives are not available, the gradient information from the surrogate model can guide exploration. In our case, the referenced approaches cannot be applied directly, as they assume a differentiable kernel. For methods that rely on a continuous latent representation (see Secs. 3.4 and 3.5), gradient information of the surrogate model in latent space can be used.

Ahmed et al. (2016) mention how several Bayesian optimization methods could leverage gradient information and encourage the community to augment their optimization schemes with gradients, supported by strong empirical results even with randomly sampled directional derivatives. Eriksson et al. (2018) alleviate the computational constraints that come from using supervised gradient information using structured kernel interpolation and computational tricks like fast matrix-vector multiplication and pivoted Cholesky preconditioning. Other avenues for mitigating the computational complexity involve using structured automatic differentiation (Ament and Gomes, 2022). Instead of using the gradient for taking stochastic steps, Penubothula et al. (2021) aim to find local critical points by querying where the predicted gradient is zero.

As mentioned above, fitting a Gaussian process to the objective allows for predicting gradients without having seen them *a priori* (Rasmussen and Williams, 2006, Sec 9.4); Müller et al. (2021) propose *Gradient Information with BO* (GIBO), in which they guide local policy search in reinforcement learning tasks, exploiting this property. Nguyen et al. (2022) address that expected gradients may not lead to the best performing outputs and compute the *most probable descent direction*.

## 3.7 Structured spaces

Some applications work over structured spaces. For example, the angles of robot arms and protein backbones map to Riemannian manifolds (Jaquier et al., 2020; Penner, 2022), and input spaces might also contain mixed variables (i.e. products of real and categorical spaces). To compute non-linear embeddings (see Sec. 3.5) followed by standard Bayesian optimization (or small variations thereof) can allow us to work over such spaces. Jaquier et al. (2020) use kernels defined on Riemannian manifolds (Feragen et al., 2015; Borovitskiy et al., 2020) and optimize the acquisition function using tools from Riemannian optimization (Boumal, 2023). The authors expand their framework to high-dimensional manifolds by projecting to lower-dimensional submanifolds, which is roughly the equivalent to *linear embeddings* in the Riemannian settings (Jaquier and Rozo, 2020).

In the categorical and mixed-variable setting, kernels over string spaces (Lodhi et al., 2000; Shervashidze et al., 2011), can be applied to BO (Moss et al., 2020). Other methods construct combinatorial graph and diffusion kernels-based GPs (Oh et al., 2019). Deshwal and Doppa (2021) combine latent space kernels with combinatorial kernels in an autoencoder-based approach.

Recently, Daulton et al. (2022b) have proposed a continuous relaxation of the discrete variables to ease the optimization of the acquisition function. Deshwal et al. (2023) propose another way to map discrete variables to continuous space, relying on Hamming distances to make a dictionary for embeddings. Papenmeier et al. (2024) extend previous work to both continuous and categorical variables: BAxUS learns an increasing sequence of subspaces using hash matrices which, when combined with the CoCaBo kernel (Ru et al., 2020), renders an algorithm for the mixed-variable setting. Finally, through a continuous relaxation of the objective that incorporates *prior* pretrained models, Michael et al. (2024) propose a surrogate on the probability vector space to optimize either the discrete input space or a continuous latent one.

**Other.** Some methods evade our taxonomy but are worth mentioning: some focus on the optimization of the acquisition function and the impact of initializations (Zhao et al., 2024; Ngo et al., 2024). Other methods balance both active learning (i.e. building a better surrogate model) and optimization (Hvarfner et al., 2023). Most recently, two articles claimed that the standard setting for Bayesian optimization or slight variations of it perform as well as the state-of-the-art of all the aforementioned families (Hvarfner et al., 2024; Xu and Zhe, 2024) – begging the question, can these methods optimize in high dimensional discrete problem spaces in a sample efficient manner?

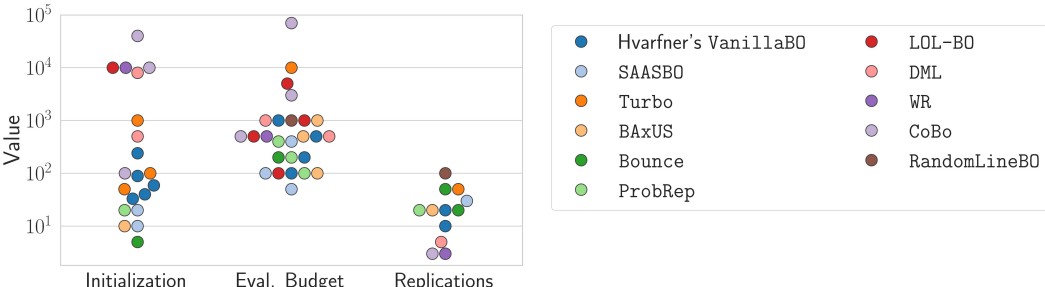

Figure 3: Initialization, evaluation budget, and nr. of replications using different seeds reported in the experimental set-ups of several HDBO methods. We see heterogeneity in the evaluation of optimizers.

## 4    Benchmarking the performance of HDBO methods

Practitioners that decide on what Bayesian optimization algorithm to use for their application will face several challenges. While surveying the field, we noticed two key discrepancies in the reported experimental set-ups: (i) the initialization varies from as low as *none* randomly/SOBOL sampled points to over $10^3$, (ii) evaluation budgets also vary for the same types of tasks. Fig. 3 visualizes these different experimental set-ups as swarmplots. Moreover, our survey covered code availability. The state-of-the-art is being pulled by workhorses, which have democratized access to GP and BO implementations: GPyTorch (Gardner et al., 2018) and BoTorch (Balandat et al., 2020), and GPFlow (Matthews et al., 2017; van der Wilk et al., 2020) and Trieste (Picheny et al., 2023). These libraries are highly useful and impactful, yet one can obtain cross-dependency conflicts between them especially if third-party dependencies are introduced or if very specific versions are required for solver setups. As a particular example, solvers like ProbRep cannot co-exist with Ax-based solvers like SAASBO or Hvarfner's VanillaBO. There is a need for *isolating* optimizers, specifying up-to-date environments in which they can run. These issues led to the development of poli.

### 4.1    poli **and** poli-baselines**: a framework for benchmarking discrete optimizers**

We want to solve truly high dimensional problems that are relevant for domains like biology and chemistry. To make the outcomes comparable, we require a unified way of defining the problem which includes consistent starting points, budgets, runtime environments, relevant assets (i.e.models used for the black box), and a logging backend invoked for every oracle observation. To that end, we implement the *Protein Objectives Library* (poli) to provide potentially isolated black box functions. Building on open source tools, poli currently provides 35 black box tasks; besides the Practical Molecular Optimization (PMO) benchmark (Huang et al., 2021; Gao et al., 2022; Brown et al., 2019), it includes Dockstring (García-Ortegón et al., 2022) as well as other protein-related black boxes like stability and solvent accessibility (Delgado et al., 2019; Blaabjerg et al., 2023; Chapman and Chang, 2000; Stanton et al., 2022).[4] The majority of

```
# pip install poli-core[ehrlich]
# pip install git+https://https://github.com/
    MachineLearningLifeScience/poli-baselines
    .git
from poli.repository import (
    EhrlichHoloProblemFactory,
)
from poli_baselines.solvers.simple.
    random_mutation import (
    RandomMutation
)

problem = EhrlichHoloProblemFactory().create(
    sequence_length=15,
    n_motifs=2,
    motif_length=7,
)
f, x0 = problem.black_box, problem.x0

solver = RandomMutation(
    black_box=f,
    x0=x0,
    y0=f(x0),
    greedy=True,   # Directed Evolution.
)

solver.solve(100)
```

black box functions can be queried with any string that complies with the corresponding alphabet making the oracles available for free-form optimization. This is an important distinction compared to pre-existing benchmarks that rely on pools of precompiled observations (Notin et al., 2023).

---

[4]A complete list can be found in our documentation: https://machinelearninglifescience.github.io/poli-docs/

| Solver\Oracle | PestControlEquiv | Ehrlich(L=5) | Ehrlich(L=15) | Ehrlich(L=64) | Sum (normalized per row) |
|---|---|---|---|---|---|
| DirectedEvolution | $0.968 \pm 0.03$ | $1.000 \pm 0.00$ | $0.448 \pm 0.16$ | $0.114 \pm 0.07$ | $2.23 \pm 0.27$ |
| HillClimbing | $0.640 \pm 0.12$ | $0.500 \pm 0.25$ | $0.392 \pm 0.20$ | $0.089 \pm 0.07$ | $0.56 \pm 0.64$ |
| CMAES | $0.816 \pm 0.06$ | $0.750 \pm 0.18$ | $0.312 \pm 0.13$ | $0.077 \pm 0.08$ | $1.24 \pm 0.45$ |
| GeneticAlgorithm | $0.712 \pm 0.02$ | $0.950 \pm 0.11$ | $0.336 \pm 0.10$ | $0.083 \pm 0.08$ | $1.50 \pm 0.31$ |
| Hvarfner's VanillaBO | $0.928 \pm 0.08$ | $0.650 \pm 0.14$ | $0.328 \pm 0.18$ | $0.079 \pm 0.08$ | $1.26 \pm 0.47$ |
| RandomLineBO | $0.624 \pm 0.11$ | $0.700 \pm 0.27$ | $0.472 \pm 0.14$ | $0.084 \pm 0.08$ | $1.04 \pm 0.60$ |
| SAASBO | $0.792 \pm 0.05$ | $0.600 \pm 0.14$ | $0.328 \pm 0.21$ | $0.075 \pm 0.06$ | $0.92 \pm 0.46$ |
| Turbo | $0.896 \pm 0.04$ | $0.850 \pm 0.14$ | $0.480 \pm 0.12$ | $0.124 \pm 0.11$ | $1.86 \pm 0.40$ |
| BAxUS | $0.712 \pm 0.08$ | $0.550 \pm 0.11$ | $0.400 \pm 0.11$ | $0.077 \pm 0.08$ | $0.79 \pm 0.39$ |
| Bounce | $1.000 \pm 0.00$ | $0.900 \pm 0.14$ | $0.416 \pm 0.13$ | $0.076 \pm 0.06$ | $2.00 \pm 0.33$ |
| ProbRep | $0.896 \pm 0.04$ | $0.950 \pm 0.11$ | $0.328 \pm 0.08$ | $0.076 \pm 0.08$ | $1.80 \pm 0.31$ |

Table 1: Sequence design problems using Ehrlich functions. Solver's performance (measured as the average best value achieved during an optimization campaign of 300 iterations) is colored according to their closeness to the known optimal value (1.0). All problems except for Ehrlich with sequence length 64 were initialized with 10 supervised samples. The remaining one was initialized with 1000.

We further provide an interface for the solvers used for the individual optimization tasks: `poli-baselines`. Consistent, stable (and up to date) environments of individual optimizers can be found therein, as well as a standardized way to query them and solve the problems raised in the previous section. These environments and optimizers are tested weekly through GitHub actions, guaranteeing their usability. An example of how `poli` and `poli-baselines` work is provided above. A problem containing a black box, an initial input, and potentially a data package is created through problem factories/a `create` method. A solver takes as input a black box and optionally supervised data, and uses the `solve` method to run the optimization for a given number of iterations. Such an interface can accommodate any optimizer that provides a method for running the optimization for a given amount of steps, as well as optimizers that do not accept custom initializations (e.g. `Bounce` or `BAxUS`). Sec. A.4 provides an introduction to this software's technical details, including a description of the logging logic though observers that keep track of every black box call.

## 4.2 Benchmarking HDBO in discrete sequences

Using the black boxes provided in `poli`, as well as the solvers provided in `poli-baselines`, we benchmark the performance of high-dimensional Bayesian optimization solvers on discrete sequences. Such optimization can take place either at the sequence level (as solvers in the *Structured Spaces* do), or in a continuous version via either one-hot representations or learned latent spaces. Benchmarks on both fronts are presented in this paper: from sequence design tasks of varying complexity (mimicking protein engineering) in one-hot/sequence space using Ehrlich functions (Stanton et al., 2024), to latent space optimization of small molecules on the Practical Molecular Optimization (PMO) benchmark (Gao et al., 2022; Huang et al., 2021; Brown et al., 2019).[5]

### 4.2.1 Sequence design problems of varying complexity

Ehrlich functions (Stanton et al., 2024) are closed-form procedurally-generated oracles in which a certain collection of motifs needs to be satisfied in a sequence of a pre-selected length. The oracle's score is between 0 and 1, and determined by how much of the individual motifs are satisfied. The number of motifs and their length are hyperparameters specified by the user. Sec. A.5.1 provides a detailed introduction. In particular, we test on 4 different configurations: a PestControl equivalent[6] with an alphabet size of 5 and a sequence length of 25 (i.e. one motif of length 25). Further, 3 different configurations imitating protein design using the alphabet of 20 amino acids and sequence lengths of 5, 15, and 64 are tested. The motif lengths and number of motifs are (4, 1), (7, 2) and (10, 4) respectively. We initialize PestControlEquiv, and the two small Ehrlich problems with 10 supervised samples, and the latter with 1000, and all methods have an evaluation budget of 300.

We optimize these oracles with representatives from the taxonomy that are frequently tested in the HDBO literature. We select Hvarfner's `VanillaBO`, `RandomLineBO`, `Turbo`, `BAxUS`, `SAASBO`, `Bounce`, and `ProbRep`, including also baselines like `DirectedEvolution` (i.e. greedily mutating

---

[5]We emphasize that the results provided in this paper are continuously updated on our project's website.

[6]PestControl is a black box whose input are 25 categorical variables, each of which has 5 potential categories. It is a commonly tested black box in the *Structured Spaces* literature.

| Oracle | Hill Climbing | Genetic Algorithm | CMAES | Hvarfner's VanillaBO | Random LineBO | SAASBO | BAxUS | Turbo | Bounce | ProbRep |
|---|---|---|---|---|---|---|---|---|---|---|
| albuterol_similarity | 0.20±0.06 | 0.32±0.03 | 0.44±0.03 | 0.19±0.01 | 0.15±0.04 | 0.43±0.05 | 0.33±0.09 | 0.35±0.03 | 0.16±0.01 | 0.21±0.03 |
| amlodipine_mpo | 0.16±0.07 | 0.29±0.06 | 0.41±0.01 | 0.12±0.07 | 0.02±0.03 | 0.41±0.02 | 0.41±0.02 | 0.42±0.03 | 0.00±0.00 | 0.00±0.00 |
| celecoxib_rediscovery | 0.04±0.02 | 0.08±0.00 | 0.16±0.01 | 0.05±0.01 | 0.02±0.00 | 0.18±0.04 | 0.16±0.01 | 0.16±0.05 | 0.02±0.01 | 0.02±0.00 |
| deco_hop | 0.53±0.01 | 0.52±0.00 | 0.54±0.01 | 0.53±0.00 | 0.52±0.00 | 0.53±0.01 | 0.53±0.02 | 0.53±0.02 | 0.50±0.00 | 0.51±0.00 |
| drd2_docking | 0.03±0.00 | 0.03±0.00 | 0.03±0.00 | 0.03±0.00 | 0.02±0.01 | 0.03±0.00 | 0.03±0.00 | 0.03±0.00 | 0.01±0.00 | 0.01±0.00 |
| fexofenadine_mpo | 0.37±0.15 | 0.48±0.03 | 0.66±0.01 | 0.40±0.03 | 0.27±0.02 | 0.64±0.04 | 0.49±0.30 | 0.54±0.24 | 0.13±0.13 | 0.20±0.08 |
| gsk3_beta | 0.23±0.11 | 0.19±0.01 | 0.20±0.04 | 0.27±0.03 | 0.20±0.07 | 0.15±0.03 | 0.12±0.04 | 0.13±0.05 | 0.09±0.08 | 0.12±0.02 |
| isomer_c7h8n2o2 | 0.63±0.15 | 0.72±0.09 | 0.73±0.12 | 0.49±0.04 | 0.03±0.15 | 0.66±0.18 | 0.49±0.05 | 0.20±0.17 | 0.11±0.09 | 0.24±0.11 |
| isomer_c9h10n2o2pf2cl | 0.48±0.16 | 0.56±0.05 | 0.57±0.14 | 0.56±0.04 | 0.32±0.21 | 0.43±0.19 | 0.36±0.16 | 0.37±0.22 | 0.01±0.01 | 0.06±0.10 |
| jnk3 | 0.19±0.03 | 0.10±0.01 | 0.10±0.03 | 0.16±0.04 | 0.10±0.03 | 0.08±0.02 | 0.08±0.02 | 0.09±0.05 | 0.05±0.04 | 0.06±0.01 |
| median_1 | 0.05±0.03 | 0.18±0.03 | 0.17±0.01 | 0.05±0.01 | 0.02±0.02 | 0.16±0.01 | 0.12±0.01 | 0.14±0.02 | 0.03±0.01 | 0.02±0.00 |
| median_2 | 0.02±0.01 | 0.08±0.01 | 0.12±0.00 | 0.03±0.01 | 0.01±0.00 | 0.12±0.00 | 0.12±0.01 | 0.12±0.02 | 0.01±0.00 | 0.01±0.00 |
| mestranol_similarity | 0.10±0.09 | 0.26±0.00 | 0.38±0.02 | 0.18±0.03 | 0.06±0.03 | 0.35±0.05 | 0.34±0.05 | 0.30±0.02 | 0.01±0.00 | 0.02±0.00 |
| osimetrinib_mpo | 0.62±0.01 | 0.63±0.01 | 0.51±0.30 | 0.60±0.01 | 0.59±0.01 | 0.59±0.06 | 0.22±0.35 | 0.33±0.32 | 0.30±0.31 | 0.59±0.04 |
| perindopril_mpo | 0.00±0.00 | 0.10±0.03 | 0.25±0.10 | 0.00±0.00 | 0.02±0.03 | 0.25±0.08 | 0.24±0.14 | 0.22±0.13 | 0.00±0.00 | 0.00±0.00 |
| ranolazine_mpo | 0.31±0.22 | 0.53±0.02 | 0.59±0.03 | 0.26±0.15 | 0.07±0.01 | 0.63±0.01 | 0.54±0.17 | 0.48±0.19 | 0.00±0.00 | 0.11±0.02 |
| rdkit_logp | 5.00±1.85 | 13.28±0.41 | 21.62±0.12 | 5.93±1.19 | 5.60±4.35 | 19.87±1.21 | 17.84±2.53 | 20.87±1.85 | 3.12±1.20 | 5.49±3.01 |
| rdkit_qed | 0.55±0.03 | 0.66±0.07 | 0.90±0.04 | 0.61±0.02 | 0.42±0.03 | 0.79±0.13 | 0.80±0.09 | 0.75±0.13 | 0.52±0.09 | 0.60±0.05 |
| sa_tdc | 8.70±0.22 | 8.71±0.20 | 7.48±0.31 | 8.69±0.10 | 8.17±0.83 | 7.14±0.88 | 7.56±0.05 | 5.55±0.31 | 8.36±0.46 | 8.59±0.13 |
| scaffold_hop | 0.37±0.01 | 0.36±0.00 | 0.39±0.01 | 0.38±0.01 | 0.37±0.03 | 0.38±0.01 | 0.37±0.01 | 0.37±0.02 | 0.34±0.01 | 0.34±0.00 |
| sitagliptin_mpo | 0.10±0.11 | 0.05±0.05 | 0.15±0.15 | 0.12±0.13 | 0.05±0.05 | 0.11±0.12 | 0.09±0.07 | 0.16±0.06 | 0.00±0.00 | 0.00±0.00 |
| thiothixene_rediscovery | 0.03±0.02 | 0.13±0.03 | 0.20±0.05 | 0.05±0.01 | 0.02±0.00 | 0.22±0.02 | 0.19±0.04 | 0.20±0.03 | 0.02±0.01 | 0.03±0.01 |
| troglitazone_rediscovery | 0.04±0.03 | 0.11±0.02 | 0.16±0.01 | 0.05±0.01 | 0.02±0.00 | 0.16±0.01 | 0.13±0.01 | 0.15±0.01 | 0.02±0.01 | 0.02±0.00 |
| valsartan_smarts | 0.00±0.00 | 0.00±0.00 | 0.00±0.00 | 0.00±0.00 | 0.00±0.00 | 0.00±0.00 | 0.00±0.00 | 0.00±0.00 | 0.00±0.00 | 0.00±0.00 |
| zaleplon_mpo | 0.05±0.09 | 0.05±0.00 | 0.13±0.04 | 0.09±0.03 | 0.00±0.00 | 0.33±0.08 | 0.05±0.05 | 0.13±0.11 | 0.00±0.00 | 0.00±0.00 |
| Sum (normalized per row) | 10.92±3.46 | 15.39±1.19 | 21.15±1.57 | 11.99±1.98 | 5.81±5.82 | 20.42±3.26 | 15.95±4.28 | 16.30±4.08 | 1.59±2.47 | 3.95±3.57 |

Table 2: Results on the PMO benchmark for a 128-latent space. The best output of the optimization campaign over max. 300 iterations are averaged over three runs, using a Sobol-sampled initialization of 10 latent points. The last row is computed by adding the result of min-max normalizing each row. Note that `Bounce` consistently ran out of memory in as few iterations as 40 (where the dimensionality of the ongoing subspace is increased), and `ProbRep` runs were stopped after 24 hours.

the incumbent in sequence space), `HillClimbing` (which explores the input space by taking random Gaussian steps), and evolutionary algorithms/strategies like `GeneticAlgorithm` and `CMA-ES`. Notice that, due to the nature of their implementations, `BAxUS` and `Bounce` are not initialized with the same supervised data (but rather according to their implementation: SOBOL sampling in a linear subspace). In this scenario, the continuous solvers are optimizing over one-hot inputs, and the discrete ones work over sequence space. The next section explores latent space optimization.

Table 1 shows the mean best value achieved in the aforementioned 4 problems with one standard deviation for 5 seeds. Problems of low complexity (i.e. PestControlEquiv and Ehrlich with seq. length 5) are readily solvable using one-hot BO, with `Bounce` solving PestControlEquiv in all 5 replications. Such problems are also easily solvable by naïve baselines like `DirectedEvolution`. For more complex problems, BO optimizers perform equally or worse than greedy baselines. We hypothesize that naïve baselines will perform worse on black boxes with higher-order effects. Note that `SAASBO` and `ProbRep` ran out of memory when instantiating the largest problem.

### 4.2.2 Benchmarking HDBO on PMO

To benchmark the performance of HDBO on discrete sequences in continuous learned representations, we consider the PMO benchmark (Gao et al., 2022; Huang et al., 2021; Brown et al., 2019).[7] We select Hvarfner's `VanillaBO`, `RandomLineBO`, `Turbo`, `BAxUS`, `SAASBO`, `Bounce`, and `ProbRep`, including also `HillClimbing`, `CMA-ES` and `GeneticAlgorithm` as baselines. All solvers start from the same initial data to ensure a fair comparison except for `BAxUS` and `Bounce` due to the nature of their implementations. We test the aforementioned methods on PMO (Gao et al., 2022; Huang et al., 2021), which requires a discrete representation of small molecules. Thus, we train two MLP VAEs on SELFIES representations of small molecules using Zinc250k (Irwin et al., 2020; Zhu et al., 2022). These generative models had 2 and 128 latent dimensions, allowing us to get an impression of how these models scale with dimensionality. We restrict sequences to be of length 70 (adding `[nop]` tokens for padding); the post-processing renders an alphabet of 64 SELFIES tokens. Details can be found in Sec. A.3.

The average best result over 3 runs (of maximum 300 iterations each) is presented in Tables 2 and 3 for 128D and 2D latent spaces respectively. We see a clear advantage in the optimizers that work on learned representations, instead of in discrete space. Such a discrepancy is to be expected: methods that optimize in latent space have been presented with information prior to their optimization campaigns, while methods like `Bounce` and `ProbRep` explore the whole discrete space. Further, the

---

[7] An introduction to PMO can be found in appendix A.5.2

| Oracle | Hill Climbing | Genetic Algorithm | CMAES | Hvarfner's VanillaBO | Random LineBO | SAASBO | Turbo | Bounce | ProbRep |
|---|---|---|---|---|---|---|---|---|---|
| albuterol_similarity | 0.31±0.10 | 0.26±0.01 | 0.38±0.05 | 0.40±0.02 | 0.36±0.02 | 0.36±0.07 | 0.40±0.08 | 0.17±0.02 | 0.21±0.03 |
| amlodipine_mpo | 0.30±0.04 | 0.30±0.05 | 0.36±0.04 | 0.34±0.02 | 0.40±0.05 | 0.39±0.06 | 0.37±0.03 | 0.00±0.00 | 0.00±0.00 |
| celecoxib_rediscovery | 0.09±0.01 | 0.06±0.02 | 0.13±0.04 | 0.13±0.01 | 0.14±0.00 | 0.14±0.01 | 0.13±0.04 | 0.02±0.01 | 0.02±0.00 |
| deco_hop | 0.51±0.00 | 0.52±0.01 | 0.52±0.01 | 0.53±0.01 | 0.53±0.01 | 0.52±0.00 | 0.53±0.01 | 0.50±0.00 | 0.51±0.00 |
| drd2_docking | 0.03±0.00 | 0.03±0.00 | 0.03±0.00 | 0.03±0.01 | 0.03±0.00 | 0.02±0.01 | 0.03±0.00 | 0.01±0.00 | 0.01±0.00 |
| fexofenadine_mpo | 0.01±0.01 | 0.43±0.02 | 0.55±0.01 | 0.60±0.04 | 0.57±0.02 | 0.59±0.07 | 0.23±0.30 | 0.13±0.13 | 0.20±0.08 |
| gsk3_beta | 0.07±0.02 | 0.19±0.01 | 0.16±0.01 | 0.14±0.02 | 0.20±0.05 | 0.18±0.05 | 0.09±0.03 | 0.09±0.08 | 0.12±0.02 |
| isomer_c7h8n2o2 | 0.45±0.05 | 0.63±0.08 | 0.42±0.10 | 0.81±0.07 | 0.79±0.10 | 0.84±0.09 | 0.25±0.13 | 0.11±0.09 | 0.24±0.11 |
| isomer_c9h10n2o2pf2cl | 0.30±0.14 | 0.46±0.17 | 0.41±0.20 | 0.49±0.01 | 0.58±0.02 | 0.52±0.18 | 0.27±0.08 | 0.01±0.01 | 0.06±0.03 |
| jnk3 | 0.10±0.03 | 0.10±0.02 | 0.06±0.01 | 0.10±0.03 | 0.08±0.02 | 0.08±0.04 | 0.06±0.02 | 0.05±0.04 | 0.06±0.01 |
| median_1 | 0.13±0.03 | 0.13±0.00 | 0.17±0.02 | 0.13±0.02 | 0.16±0.01 | 0.15±0.04 | 0.17±0.03 | 0.03±0.01 | 0.02±0.00 |
| median_2 | 0.08±0.01 | 0.08±0.02 | 0.11±0.01 | 0.11±0.01 | 0.12±0.02 | 0.12±0.01 | 0.10±0.01 | 0.01±0.00 | 0.01±0.00 |
| mestranol_similarity | 0.37±0.01 | 0.28±0.03 | 0.36±0.05 | 0.34±0.00 | 0.35±0.03 | 0.36±0.03 | 0.32±0.09 | 0.01±0.00 | 0.02±0.00 |
| osimetrinib_mpo | 0.00±0.00 | 0.62±0.01 | 0.34±0.28 | 0.65±0.03 | 0.65±0.06 | 0.64±0.00 | 0.45±0.39 | 0.30±0.31 | 0.59±0.04 |
| perindopril_mpo | 0.12±0.01 | 0.10±0.08 | 0.18±0.04 | 0.14±0.00 | 0.14±0.01 | 0.23±0.10 | 0.20±0.08 | 0.00±0.00 | 0.00±0.00 |
| ranolazine_mpo | 0.58±0.09 | 0.54±0.06 | 0.57±0.09 | 0.62±0.13 | 0.69±0.08 | 0.55±0.07 | 0.29±0.05 | 0.00±0.00 | 0.11±0.02 |
| rdkit_logp | 20.85±0.21 | 14.76±0.45 | 20.67±0.66 | 20.73±1.05 | 18.74±1.83 | 16.94±7.17 | 20.15±3.07 | 3.12±1.20 | 5.49±3.01 |
| rdkit_qed | 0.82±0.06 | 0.56±0.02 | 0.82±0.11 | 0.77±0.02 | 0.77±0.10 | 0.80±0.10 | 0.63±0.03 | 0.52±0.09 | 0.60±0.05 |
| sa_tdc | 7.64±0.47 | 8.74±0.11 | 7.00±0.68 | 7.98±0.02 | 6.86±1.96 | 7.96±0.05 | 7.16±0.30 | 8.36±0.46 | 8.59±0.13 |
| scaffold_hop | 0.34±0.00 | 0.36±0.01 | 0.37±0.00 | 0.37±0.00 | 0.37±0.00 | 0.37±0.01 | 0.38±0.01 | 0.34±0.01 | 0.34±0.00 |
| sitagliptin_mpo | 0.05±0.03 | 0.02±0.01 | 0.08±0.07 | 0.08±0.06 | 0.08±0.06 | 0.15±0.13 | 0.00±0.01 | 0.00±0.00 | 0.00±0.00 |
| thiothixene_rediscovery | 0.15±0.02 | 0.13±0.03 | 0.18±0.02 | 0.21±0.04 | 0.18±0.05 | 0.18±0.03 | 0.15±0.01 | 0.02±0.01 | 0.03±0.01 |
| troglitazone_rediscovery | 0.10±0.01 | 0.11±0.02 | 0.17±0.02 | 0.15±0.00 | 0.16±0.01 | 0.14±0.01 | 0.12±0.02 | 0.02±0.01 | 0.02±0.00 |
| valsartan_smarts | 0.00±0.00 | 0.00±0.00 | 0.00±0.00 | 0.00±0.00 | 0.00±0.00 | 0.00±0.00 | 0.00±0.00 | 0.00±0.00 | 0.00±0.00 |
| zaleplon_mpo | 0.02±0.02 | 0.02±0.01 | 0.12±0.05 | 0.15±0.10 | 0.22±0.11 | 0.12±0.04 | 0.05±0.04 | 0.00±0.00 | 0.00±0.00 |
| Sum (normalized per row) | 12.35±1.33 | 13.74±1.26 | 17.28±2.56 | 19.54±1.81 | 19.54±4.59 | 19.24±8.38 | 14.52±4.84 | 1.65±2.47 | 3.87±3.57 |

Table 3: Results on the PMO benchmark for a 2-dimensional latent space. The best output of the optimization campaign over max. 100 iterations are averaged over three runs, using 10 initial SOBOL-sampled points. The last row is computed as in 2. `Bounce` and `ProbRep`'s underlying results are exactly the same as in the 128D case, but restricted to 100 iterations. We note that `BAxUS` defaults to `Turbo` in 2 dimensions.

simple baseline is reliably beaten by the continuous alternatives in lower dimensions except `Turbo`, but this advantage is not as clear in the 128D case, signaling a more complex problem. Some of these tasks, however, are equally challenging for all solvers. `deco_hop` remains close to the original default value of 0.5, and there is no improvement over `valsartan_smarts` (which only REINVENT improves on in the original PMO results (Gao et al., 2022; Loeffler et al., 2024)).

## 5 Conclusion

In this paper, we surveyed the field of high-dimensional Bayesian optimization (HDBO) focusing on discrete problems. This highlighted the need for (i) a novel taxonomy of the field that emphasizes the differences between methods that rely on unsupervised discrete information, and methods that optimize sequences directly, and (ii) a standardized framework for benchmarking HDBO methods. We approach these in the form of two software tools: `poli` and `poli-baselines`. Using these tools, we implemented several HDBO methods and tested them in a standard benchmark for sequence design and small molecule optimization. Our findings suggest that optimizers that work on discrete sequence space (i.e. the *structured spaces* family) perform as expected on problems of low complexity, but do not scale gracefully to problems with larger dictionaries/higher sequence lengths. In such cases, optimizers that leverage pre-trained latent-variable models have an edge over the other tested methods that work directly on sequence space. That being said, we find that, for low budgets, simpler baselines tend to perform as well or better than most BO methods, echoing recent criticisms of molecular optimization (Tripp and Hernández-Lobato, 2023). Our framework opens the door to fair and easily-replicable comparisons. We expect `poli-baselines` to be used by practitioners for running HDBO solvers in up-to-date environments compared across several tasks in our ongoing benchmark, which we plan to expand to other discrete objectives in `poli`.

**Limitations and societal impact.** Although we taxonomize different families of HDBO methods, we have only benchmarked a subset; moreover, the black boxes we use are often simple, closed-form functions or data-driven oracles, limiting the applicability of our benchmark to real-world scenarios. The field of discrete sequence optimization for biology/chemistry needs better black box oracles with more relevance to their respective domains. Another limitation of our approach is the lack of a systematic analysis on the myriad of design choices made on these optimization campaigns; we strived for documenting them properly, but an in-depth analysis is called for in future work. Our benchmark is ongoing and we plan to include further experiments in the project's website with, hopefully, participation from the community. Finally, we note that optimizing small molecules or proteins opens the door to both drug discovery, but also dual use (Urbina et al., 2022).

## Acknowledgments and Disclosure of Funding

The work was partly funded by the Novo Nordisk Foundation through the Center for Basic Machine Learning Research in Life Science (NNF20OC0062606). RM is funded by the Danish Data Science Academy, which is funded by the Novo Nordisk Foundation (NNF21SA0069429) and VILLUM FONDEN (40516). SH was further supported by a research grant (42062) from VILLUM FONDEN as well as funding from the European Research Council (ERC) under the European Union's Horizon programme (grant agreement 101125993). WB was supported by VILLUM FONDEN (40578). This work was in part supported by the Pioneer Centre for AI (DRNF grant number P1). MGD thanks Sergio Garrido and Anshuk Uppal for feedback on early versions of this document; Peter Mørch Groth for useful discussions; Luigi Nardi, Erik Hellsten and Raffaello Baluyot for feedback on the preprint version of this paper, and Samuel Stanton for feedback on the software.

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

# A Appendix

## A.1 Methods Overview

| Method | Date (first occurrence) | Reference | Code Available |
|---|---|---|---|
| SOLID | January 23, 2019 | Winkel et al. (2021) | ✗ |
| Deep GPs | May 7, 2019 | Hebbal et al. (2021) | ✗ |
| ASM | July 15, 2017 | Palar and Shimoyama (2017) | ✗ |
| Add-GP-UCB | May 13, 2015 | Han et al. (2021) | ✓ |
| TuRBO | December 8, 2019 | Eriksson et al. (2019) | ✓ |
| LOL-BO | January 28, 2022 | Maus et al. (2022) | ✓ |
| ROBOT | October 20, 2022 | Maus et al. (2023) | ✓ |
| REMBO | January 9, 2013 | Wang et al. (2016) | ✓ |
| SAASBO | June 10, 2021 | Eriksson and Jankowiak (2021) | ✓ |
| Dropout | August 19, 2017 | Li et al. (2018) | ✗ |
| BAxUS | November 28, 2022 | Papenmeier et al. (2022) | ✓ |
| LineBO | June 10, 2019 | Kirschner et al. (2019) | ✓ |
| ALEBO | December 6, 2020 | Letham et al. (2020) | ✓ |
| HeSBO | May 24, 2019 | Nayebi et al. (2019) | ✓ |
| BORING | October 5, 2020 | Yenicelik (2020) | ✗ |
| REMBO 2.0 | October 18, 2019 | Binois et al. (2020) | ✗ |
| Quantile-GP BO | February 1, 2020 | Moriconi et al. (2020b) | ✗ |
| Warped REMBO | January 1, 2015 | Binois et al. (2015) | ✗ |
| closed-form ASM | September 22, 2020 | Wycoff et al. (2021) | ✓ |
| Active manifolds | May 24, 2019 | Bridges et al. (2019) | ✗ |
| Deep GPs (MO) | January 1, 2019 | Hebbal et al. (2019) | ✗ |
| LADDER | December 6, 2021 | Deshwal and Doppa (2021) | ✓ |
| Attr. Adjustment | August 6, 2018 | Eissman et al. (2018) | ✗ |
| VAEs DML | November 1, 2021 | Grosnit et al. (2021) | ✓ |
| LSBO | February 28, 2018 | Gómez-Bombarelli et al. (2018) | ✓ |
| Weigh. Retraining | October 25, 2020 | Tripp et al. (2020) | ✓ |
| MORBO (MO) | September 22, 2021 | Daulton et al. (2022a) | ✓ |
| TREGO | October 10, 2022 | Diouane et al. (2023) | ✗ |
| TRIKE | August 8, 2015 | Regis (2016) | ✗ |
| TRPBO | November 21, 2020 | Zhou et al. (2021) | ✗ |
| D-SKIP | December 3, 2018 | Eriksson et al. (2018) | ✓ |
| RDUCB | May 29, 2023 | Ziomek and Bou Ammar (2023) | ✓ |
| G-Add-GP-UCB | April 9, 2018 | Rolland et al. (2018) | ✗ |
| QFF | December 3, 2018 | Mutny and Krause (2018) | ✗ |
| SI-BO | December 5, 2013 | Djolonga et al. (2013) | ✗ |
| SRE-IMGPO | July 9, 2016 | Qian et al. (2016) | ✗ |
| HDS | June 27, 2012 | Chen et al. (2012) | ✗ |
| AL of LEs | October 24, 2013 | Garnett et al. (2013) | ✓ |
| SG-VAE | July 3, 2018 | Lu et al. (2018) | ✗ |
| BO+MCMC | April 10, 2017 | Gardner et al. (2017) | ✓ |
| Ensemble BO | March 31, 2018 | Wang et al. (2018) | ✓ |
| BOCK | March 3, 2018 | Oh et al. (2018) | ✓ |
| COMBO | December 8, 2019 | Oh et al. (2019) | ✓ |
| MGPC-BO | September 1, 2020 | Moriconi et al. (2020a) | ✓ |
| G-STAR | December 1, 2016 | Pedrielli and Ng (2016) | ✗ |
| CASMOPOLITAN | June 18, 2021 | Wan et al. (2021) | ✓ |
| Vanilla BO | February 25, 2024 | Hvarfner et al. (2024) | ✓ |
| Bounce | July 2, 2023 | Papenmeier et al. (2024) | ✓ |
| EGP | August 7, 2017 | Rana et al. (2017) | ✗ |
| CoBO | December 10, 2023 | Lee et al. (2023) | ✓ |
| DSO | February 27, 2024 | Kong et al. (2024) | ✗ |
| MPD | January 16, 2023 | Nguyen et al. (2022) | ✓ |

| | | | |
|---|---|---|---|
| GIBO | November 22, 2021 | Müller et al. (2021) | ✓ |
| PR | October 18, 2022 | Daulton et al. (2022b) | ✓ |
| RPP | May 9, 2016 | Li et al. (2016) | ✗ |
| MCTS-VS | October 31, 2022 | Song et al. (2022) | ✓ |
| Standard BO | February 5, 2024 | Xu and Zhe (2024) | ✓ |
| Scalable FOBO | June 16, 2022 | Ament and Gomes (2022) | ✓ |
| FOBO | December 6, 2017 | Ahmed et al. (2016) | ✗ |
| BOSS | October 2, 2020 | Moss et al. (2020) | ✓ |
| SILBO | May 29, 2020 | Chen et al. (2020) | ✓ |
| SCoreBO | April 21, 2023 | Hvarfner et al. (2023) | ✓ |
| HEBO | December 7, 2020 | Cowen-Rivers et al. (2022) | ✓ |
| Tree Add-GP-UCB | May 1, 2021 | Han et al. (2021) | ✓ |
| SIR/SDR | July 21, 2019 | Zhang et al. (2019) | ✓ |
| GaBO | November 22, 2021 | Jaquier et al. (2020) | ✓ |
| ECI | April 18, 2024 | Zhan (2024) | ✓ |
| MAVE-BO | March 8, 2024 | Hu et al. (2024) | ✗ |
| CMA-BO | February 5, 2024 | Ngo et al. (2024) | ✓ |
| RTDK-BO | October 5, 2023 | Shmakov et al. (2023) | ✗ |
| PG-LBO | December 28, 2023 | Chen et al. (2024) | ✓ |
| TSBO | May 4, 2023 | Yin et al. (2024) | ✗ |
| BODi | March 3, 2023 | Deshwal et al. (2023) | ✓ |
| Mahalanobis BatchBO | November 2, 2022 | Horiguchi et al. (2022) | ✓ |
| KPCA-BO | April 28, 2022 | Antonov et al. (2022) | ✓ |
| PCA-BO | July 2, 2020 | Raponi et al. (2020) | ✓ |
| DSA | November 18, 2015 | Ulmasov et al. (2016) | ✓ |
| RPA-GP | June 12, 2020 | Delbridge et al. (2020) | ✓ |
| HD-GaBO | December 6, 2020 | Jaquier and Rozo (2020) | ✓ |
| Amortized BO | May 27, 2020 | Swersky et al. (2020) | ✗ |
| d-KG | December 4, 2017 | Wu et al. (2017) | ✓ |
| Prabuchandran's FOBO | March 1, 2021 | Penubothula et al. (2021) | ✗ |
| AlgFOO | March 18, 2021 | Shekhar and Javidi (2021) | ✗ |
| GTBO | October 5, 2023 | Hellsten et al. (2023) | ✓ |
| CoCaBo | June 12, 2020 | Ru et al. (2020) | ✓ |
| MercBO | February 2, 2019 | Deshwal et al. (2021b) | ✓ |
| HyBO | July 18, 2021 | Deshwal et al. (2021a) | ✓ |
| BOCS | July 10, 2018 | Baptista and Poloczek (2018) | ✓ |
| LaMBO | July 22, 2022 | Stanton et al. (2022) | ✓ |
| LaMBO-2 | December 12, 2023 | Gruver et al. (2023) | ✓ |
| G.M. & Lobato | March 1, 2020 | Garrido-Merchán and Hernández-Lobato (2020) | ✗ |
| BOHAMIANN | December 5, 2016 | Springenberg et al. (2016) | ✓ |
| AIBO | February 16, 2023 | Zhao et al. (2024) | ✓ |
| CoRel | April 26, 2024 | Michael et al. (2024) | ✓ |
| VAE-BO+Inv | July 22, 2022 | Verma et al. (2022) | ✗ |
| Uncert | November 09, 2021 | Notin et al. (2021) | ✓ |
| GAUCHE | September 21, 2023 | Griffiths et al. (2023) | ✓ |
| LLMs | July 21, 2024 | Kristiadi et al. (2024) | ✓ |
| AntBO | January 23, 2023 | Khan et al. (2023) | ✓ |
| RMF | May 20, 2022 | Kim et al. (2022) | ✗ |

Table 4: References for the methods presented in the taxonomy

### A.1.1 On how HDBO can be applied to discrete sequences

Fig. 4 shows an overview of the different ways in which HDBO could be applied to discrete sequences. On the left, we have the original sequence space with inputs being a list of categorical variables belonging to some alphabet; methods like Bounce, Prob. Rep and those that work on mixed/categorial inputs in the *Structured Spaces* family (see Sec. 3.7) work in this space. Since the original sequence space is large for the problems we are interested in, several other methods rely on subspaces from the original problem. Such subspaces can be constructed either as linear projections (Sec. 3.4) or as

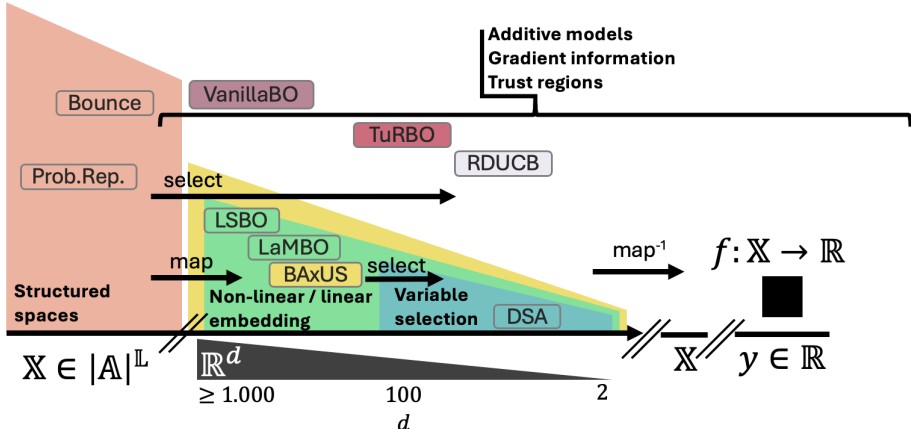

Figure 4: Overview of problem-space (x-axis) and how the categories act on the space. The black box ultimately maps from the discrete sequences of alphabet elements to a real value. The BO methods can act on the original (discrete) space, linear or non-linear mappings of it or selected variables of the input space or a mapping of it. These continuous versions can also accommodate one-hot representations.

learned non-linear latent variables (Sec. 3.5). Once a latent continuous representation is available, all other families in the taxonomy become available and could be leveraged for latent space optimization. In these cases, there is usually a mapping between the subspace and the original high-dimensional space which allows for computing objective functions. In the linear case an up-projection is given by multiplying with a matrix of the right shape, and in the case of non-linear representations it is usually a decoding process.

In this paper, we explore applying the discrete optimizers of the *Structured Spaces* family in the sequence spaces of several Ehrlich problems (Sec. 4.2.1), as well as small molecule optimization using SELFIES representations. The continuous optimizers chosen from the other families are applied directly in one-hot space in the case of Ehrlich, or in the latent space of a Variational Autoencoder learned for SELFIES strings (Sec. 4.2.2).

## A.2 Reproducing results

**Bounce.** We forked the official open source implementation of `Bounce`[8] and added an interface between `poli`'s black boxes and their optimizer. Moreover, we made their implementation pip-installable. `Bounce`'s implementation is originally provided with an MIT license.

**Probabilistic Reparametrization** provides an open-source implementation built on `GPyTorch` and `BoTorch`.[9] They provide a `pip` installable Python package which did not install until the dependencies mentioned above were fixed (to 1.11 and 0.7 respectively); further, the environment provided had to be updated by replacing the deprecated `scikit-learn` installation in a fork of their repository. After implementing an interface for `poli` black boxes, we relied on their script `run_one_replication.py` to implement a custom solver. The original PR code is provided with an MIT License.

**SAASBO, Hvarfner's Vanilla BO** were all implemented by following the tutorials in `Ax`. Ax provides models for SAASBO, and we implemented a `BoTorch` model following the original implementation. We also provide an implementation of `ALEBO` using `Ax`. Ax is provided with an MIT License, and Hvarfner's original code does not have a license in GitHub yet.

**Turbo** was implemented by following the tutorial on BoTorch,[10] which can be found on GitHub under MIT License.

---

[8]https://github.com/LeoIV/bounce
[9]https://github.com/facebookresearch/bo_pr
[10]https://github.com/pytorch/botorch/blob/main/tutorials/turbo_1.ipynb

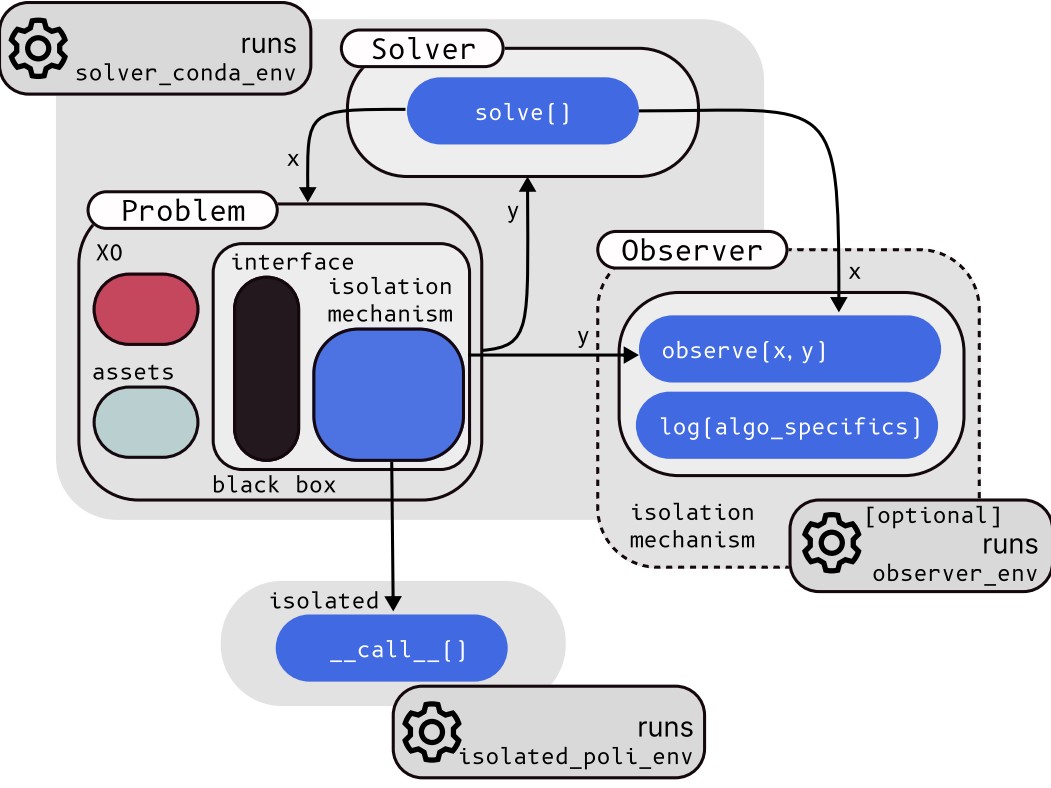

Figure 5: `poli`'s isolation process for complex environments

**BAxUS** is implemented using the Python package provided by the authors.[11] Since their work builds on the original TuRBO code from Uber, it inherits their license.

### A.3 Training VAEs on SELFIES

**Models.** We use `PyTorch` to implement a VAE with [one_hot_input, 1024, 512, 256, latent_dim] encoder and a symmetric decoder, using ReLU activations. The latent space prior used is a standard Gaussian, and the decoded distribution is a categorical. We use `torch.distributions` to compute ELBO losses without any $\beta$ weighting.[12]

**Training regimes.** We train for a maximum of 1000 epochs using early stopping with a patience of 50 epochs. Our batch sizes are 512, learning rates are $5 \times 10^{-4}$, and optimizer is AdamW.[13]

### A.4 Technical details on `poli` and `poli-baselines`

This section gives a short overview on the possibilities of `poli` and `poli-baselines`. For further information please refer to the full online documentation under https://machinelearninglifescience.github.io/poli-docs/.

---

[11] https://github.com/LeoIV/BAxUS

[12] The exact implementation can be found here: https://github.com/MachineLearningLifeScience/hdbo_benchmark/blob/1a81f7fbd531eb4dc70f84828d6efc1f2ee53e5e/src/hdbo_benchmark/generative_models/vae_selfies.py. We are currently exploring a generative model with more expressive power, and plan to update the benchmarks accordingly in our project's website.

[13] The training script can be found here: https://github.com/MachineLearningLifeScience/hdbo_benchmark/blob/master/src/hdbo_benchmark/experiments/training_vae_on_zinc250k/train_vae.py

The aim of `poli` is to make it easier to benchmark new algorithms on new problems. There are three components to our framework: **problems** created by problem factories which contain black boxes, **solvers** that optimize these problems (aiming always at maximization), and **observers** which get attached to black boxes and log every single black box call. Fig. 5 shows these three components and their relationships. `poli` provides a unified `numpy` (Harris et al., 2020) interface, with inputs being `numpy` arrays of strings, and outputs being `numpy` arrays of floats. Finally, `poli` makes sure to log every call to the black box, as well as handling evaluation budgets—**function calls are measured transparently and consistently across algorithms**.

A major issue on the endeavour of benchmarking on new problems are often conflicting package dependencies. When evaluating a brand-new solver on an older benchmark problem, it can simply be impossible to create a Python environment in which both can run. One may re-implement one or the other, though this is not only cumbersome but also error-prone. With the isolation mechanisms provided by `poli` this is no longer a problem. In particular, the isolation of the different components in the system (the black box, or the observer that is used for logging) allows an easy comparison to additional algorithms or on other problems, without the need to change plotting scripts.

Listing 1 shows the general workflow:[14] a user instantiates a black box by calling `create` with the corresponding black box's name, and the name of an observer. `poli` then takes care that both black box and observer are instantiated, in different conda environments if necessary. `create` returns an object inheriting from `Problem` which provides access to all things necessary: the black box function, initial observations, the observer and problem information like the alphabet and if sequences are aligned or not.[15] The black box is a function that takes and returns a `numpy` array, and the observer is informed automatically of any such calls. Each component of this workflow (black box, algorithm, and observer) can be easily exchanged by user-developed classes more specifically tailored to the need at hand. We refer to our online documentation[16] for the most recent information on how to do this.

To analyze an experiment a user has to define an observer, inheriting from `AbstractObserver`. Listing 2 shows how to quickly set up a simple text-file logger. The philosophy is to define the quantities of interest independent of problem or algorithm. We provide standard observers, for example an `mlflow` observer, and we encourage users to implement their own, to log other metrics of interest. Just as black boxes, observers can run in isolation, such that the environment is independent of problem and, or solver, to facilitate consistent recording of metrics. Whenever observers can run directly in the same environment as the solver, they could be attached directly to the black box using the `set_observer` method. Listing 3 shows an example using the same workflow as above, and the `SimpleObserver` presented in listing 2.

To expand on the suite of available black boxes, the user has to implement a subclass of the core object of `poli`: `AbstractBlackBox`. The method to implement is `_black_box`, which takes as input an array of strings x (as well as an optional context), and outputs the result of evaluating it as an array of floats. Listing 4 shows an example code snippet. At run time, another user wanting to test an algorithm just imports the relevant black box object from `poli`'s repository, and gets an interface to a dynamically instantiated black box potentially running in a different conda environment. The `__call__` method then takes care of communication with the caller and logging to an observer. Some of these black boxes require additional assets (*e.g.* the weights of a neural network, or `csv` files). They are all either already provided or dynamically downloaded when the black box is used.

Optimization algorithms can be oblivious to any requirements a black box might have. Solvers have only access to the `AbstractBlackBox` interface. They communicate with the problem only indirectly via a local network socket provided by `python`'s native `multiprocessing` library. This is the **isolation mechanism** that allows both algorithm and problem to run in different python environments.

---

[14]All listings in this appendix were tested using `poli` version v1.0.1, and `poli-baselines` version v1.0.2. They can be easily implemented and tested in a Colab notebook.

[15]A future release adds a *data package* attribute, attached to problems. The data package standardizes the information available to solvers by declaring unsupervised and supervised data; equalizing solver initialization.

[16]https://machinelearninglifescience.github.io/poli-docs/

```python
# the main function to instantiate problems
from poli import create

# an example solver from poli-baselines
from poli_baselines.solvers.simple.random_mutation import
                                    RandomMutation

# To create a problem, a user has to provide its name.
# In some cases, further parameters need to be specified.
# Optionally, the user can attach an observer through its
# name, or by using the problem.black_box.set_observer
# method.
problem = create(
    name="aloha",
    observer_name="",
    seed=0,
    observer_init_info={"CALLER": RandomMutation.__name__}
)

# The problem holds the black box, initial data,
# and other information.
f, x0 = problem.black_box, problem.x0

# Evaluate the initial inputs if desired.
y0 = f(x0)

# poli-baselines solvers simply take the black box ...
solver = RandomMutation(black_box=f, x0=x0, y0=y0)

# ... and then try to solve (maximize) the problem.
solver.solve(max_iter=1000)

# If desired, an algorithm can also send information to the observer.
# problem.observer.log({"FOO", "BAR"})
```

Code Listing 1: For `poli`-integrated problems and algorithms setting up an experiment is easy. This listing shows an example of running an experiment using the create method. In the main document, you can find another example that imports the problem factories directly.

## A.5  Benchmark Introduction

### A.5.1  Ehrlich functions

Ehrlich functions (Stanton et al., 2024) were proposed as a closed-form alternative to black boxes like `FoldX` or `RaSP`, which could potentially raise licensing issues and, before `poli`, were not readily available for querying. This section explains how Ehrlich functions are constructed and queried from a bird's eye view. The details can be found in the original paper.

These functions are procedurally generated from a random seed and hyperparameters like alphabet size $|\mathcal{A}|$, sequence length $L$, number of motifs $n_m$ and motif length $l$. The following are the steps that are followed to construct the problem:

1. A sparse transition matrix $A$ of size $|\mathcal{A}| \times |\mathcal{A}|$ is constructed. This transition matrix spans a finite Markov chain: sequences in the problem are constructed by starting with a random symbol and following the probabilities determined by $A$. The rows of this transition matrix are probability vectors which determine how likely it is to go from one symbol in the alphabet to another. Such a transition matrix needs to satisfy that (i) the probability of going from one symbol to itself is non-zero, and (ii) it is always possible to go from any symbol to any other symbol after following a sequence of finite steps (irreducibility). In other words, we need the Markov chain spanned by $A$ to be ergodic. $A$ defines the search space of feasible sequence: the fact that $A$ is sparse means that some of the transitions are impossible.

```python
from pathlib import Path
from uuid import uuid4
import json

import numpy as np

from poli.core.black_box_information import BlackBoxInformation
from poli.core.util.abstract_observer import AbstractObserver
from poli.core.registry import register_observer

class SimpleObserver(AbstractObserver):
    # The init and initialize_observer methods
    def initialize_observer(
        self,
        problem_setup_info: BlackBoxInformation,
        caller_info: object,
        seed: int,
    ) -> object:
        # Defining the experiment path
        self.experiment_path = caller_info["experiment_path"]

        # Saving the metadata for this experiment
        metadata = problem_setup_info.as_dict()

        # Saving the initial evaluations and seed
        metadata["seed"] = seed

        # Saving the metadata
        with open(self.experiment_path / "metadata.json", "w") as f:
            json.dump(metadata, f)

    def observe(self, x: np.ndarray, y: np.ndarray, context=None) ->
                                None:
        # Appending these results to the results file.
        with open(self.experiment_path / "results.txt", "a") as fp:
            fp.write(f"{x.tolist()}\t{y.tolist()}\n")

if __name__ == '__main__':
    # This part needs to be done only once.
    # poli notes down the location of the current conda environment,
                                so that if necessary, the observer
                                can be instantiated in isolation.
    register_observer(
        observer=SimpleObserver(), observer_name="simple_observer"
    )
```

Code Listing 2: Observing an experiment

```python
# Creating a fresh problem
problem = create(
    name="aloha",
    seed=0,
)

# Getting the black box and initial input
f, x0 = problem.black_box, problem.x0

# Initializing an observer
observer = SimpleObserver()
observer.initialize_observer(
    problem_setup_info=f.info,
    caller_info={
      "experiment_path": Path().parent,
    },
    seed=0,
)

# Setting it
f.set_observer(observer)

# Evaluate the initial inputs if desired.
y0 = f(x0)

# poli-baselines solvers simply take the black box ...
solver = RandomMutation(black_box=f, x0=x0, y0=y0)

# ... and then try to solve (maximize) the problem.
solver.solve(max_iter=1000)
```

Code Listing 3: Attaching an observer directly.

```python
from string import ascii_uppercase

import numpy as np
from poli.core.abstract_black_box import AbstractBlackBox
from poli.core.black_box_information import BlackBoxInformation

class OurAlohaBlackBox(AbstractBlackBox):
    # The only method you need to define
    def _black_box(self, x, context = None):
        matches = x == np.array(["A", "L", "O", "H", "A"])
        return np.sum(matches, axis=1, keepdims=True)

    def get_black_box_info(self) -> BlackBoxInformation:
        return our_aloha_information  # Could be dynamic

our_aloha_information = BlackBoxInformation(
    name="our_aloha",
    max_sequence_length=5,
    aligned=True,
    fixed_length=True,
    deterministic=True,
    alphabet=list(ascii_uppercase),
    discrete=True,
)

f = OurAlohaBlackBox()
f(np.array([["A", "L", "O", "O", "F"]]))  # returns [[3]]
```

Code Listing 4: Implementing a black box

2. Once $A$ has been constructed, $n_m$ motifs of length $l$ are sampled according to it by sampling a sequence of length $n_m \cdot l$ and splitting it. This ensures that the motifs are feasible and easily within reach from one to the next.

3. These motifs are to be satisfied in certain positions in the string. These positions are constructed using random offsets of size $l$. For example, the motif "ADER" with offsets of $[0, 1, 3, 5]$ is satisfied if the sequence contains "ADXEXR" where "X" can be any other member of the alphabet: "D" is at distance 1 of "A", "E" is at a distance 3 of "A" and "R" is at distance 5 of "A". These random offsets are randomly constructed to maximize slackness in the sequence.

We take the original implementation[17] and wrap our black box logic around it, making it compatible with solvers in `poli-baselines`. We also include a data package made by using the original utilities for sampling the underlying transition matrix.

### A.5.2 PMO

The *Practical Molecular Optimization* (PMO) benchmark contains a representative set of tasks defined on small molecule inputs with respective computational oracles. The input to the black box functions are alphabet representation for small molecules as either tokenized SMILES (Weininger, 1988) or SELFIES (Krenn et al., 2020) (see below) for the exact alphabets used – in principle one can be converted into the other. While *small* molecule sequences usually contain fewer elements in a sequence than for example proteins, the alphabet can contain more tokens making this yet another high dimensional discrete optimization space. We build upon the work by Gao et al. (2022), who propose molecular optimization focused on validity, diversity, synthesizability - using computational values for all metrics and a budget of 10.000 evaluations. The PMO suite itself is based on the existing benchmarks Guacamol (Brown et al., 2019), and elements of the TDC (Huang et al., 2021). The types of optimization tasks for small molecules can be differentiated into: optimizing for simple metrics like QED, LogP[18] qualitative tasks, such as scaffold hopping, rediscovery of particular substances (e.g. troglitazone), and more complex tasks such as docking surrogates and classifiers on fingerprint representations of molecules (GSK3, JNK3, DRD2) (Huang et al., 2021). We evaluate a selection of solvers across all tasks with multiple seeded runs on a fixed budget and report the average of the best observations, weighing all functions equally.[19] Altogether, these tasks constitute a set of functions which can be optimized and which have previously been used to assess algorithm performance in the bio-chemical domain, allowing a comparisons to previous results and reported benchmarks. We make these environments accessible and solvable in a unified way through the `poli` infrastructure.

```
{"[nop]": 0, "[C]": 1, "[=C]": 2, "[Ring1]": 3, "[Branch1]": 4, "[N]": 5, "[=Branch1]
": 6, "[=O]": 7, "[O]": 8, "[Branch2]": 9, "[Ring2]": 10, "[=N]": 11, "[S]": 12, "[#
Branch1]": 13, "[C@@H1]": 14, "[C@H1]": 15, "[=Branch2]": 16, "[F]": 17, "[#Branch2]
": 18, "[Cl]": 19, "[#C]": 20, "[NH1+1]": 21, "[P]": 22, "[O-1]": 23, "[NH2+1]": 24,
 "[Br]": 25, "[N+1]": 26, "[#N]": 27, "[C@]": 28, "[NH3+1]": 29, "[C@@]": 30, "[=S]"
: 31, "[=NH1+1]": 32, "[N-1]": 33, "[=N+1]": 34, "[S@]": 35, "[S@@]": 36, "[I]": 37,
 "[S-1]": 38, "[=NH2+1]": 39, "[=S@@]": 40, "[=S@]": 41, "[=N-1]": 42, "[P@@]": 43,
"[P@]": 44, "[NH1-1]": 45, "[=O+1]": 46, "[=P]": 47, "[=P@@]": 48, "[=OH1+1]": 49, "
[=P@]": 50, "[#N+1]": 51, "[S+1]": 52, "[CH1-1]": 53, "[=SH1+1]": 54, "[P@@H1]": 55,
 "[=PH2]": 56, "[P+1]": 57, "[CH2-1]": 58, "[O+1]": 59, "[=S+1]": 60, "[PH1+1]": 61,
 "[PH1]": 62, "[S@@+1]": 63}
```
Code Listing 5: Tokenized SELFIES alphabet used. Size=64 tokens, maximum length=70.

## A.6 Compute details for all experiments

### A.6.1 Ehrlich functions

The following solvers ran in an HPC with memory/CPU/GPU requirements given by 30G/10/1 Titan X with 12Gb of memory with: `SAASBO`, `ProbRep`, Hvarfner's `VanillaBO`, `RandomLineBO`, `DirectedEvolution`, `HillClimbing`, `CMA-ES` and `GeneticAlgorithm`. All these were time-gated via SLURM to run for at most 24hrs.

---

[17]https://github.com/prescient-design/holo-bench

[18]These are aggregate properties of small molecules, and can sometimes be poor proxies for other chemical downstream behavior.

[19]Equal weighting can favor methods that perform well on a few exploitable tasks (e.g. logP, QED). We draw the users attention to that fact with the alternatives to discount or discard such tasks.

The remaining solvers (`BAxUS`, `Turbo` and `Bounce`) ran in a Google Cloud instance with one Tesla T4 with 16Gb.

### A.6.2 PMO

`HillClimbing`, Hvarfner's `VanillaBO`, `RandomLineBO`, `SAASBO` and `ProbRep` experiments ran in an HPC cluster on CPUs using equivalent SLURM scripts (max 24h of runtime).

`Turbo` ran on an M2 Max Mac with 32Gb of memory using MPS.

`BAxUS` and `Bounce` ran on a Deep Learning compute server using the marketplace solutions of Google Cloud Platform, using a Tesla T4 (approx. 16Gb of memory).

