# OpenReview forum: "A survey and benchmark of high-dimensional Bayesian optimization of discrete sequences"
_NeurIPS.cc/2024/Datasets_and_Benchmarks_Track — NeurIPS 2024 Track Datasets and Benchmarks Poster_

### Official Review · Reviewer_1TmH · 2024-07-19
**Nice survey, however the benchmarking is underwhelming and there is litte-to-no practical description of the software libraries introduced**

**Rating:** 5
**Confidence:** 3

**Review:**

The paper is clear and reads well. It adds to the growing literature that attempts to unify and provide a practical taxonomy for HDBO practitioners.
I do not believe this submission goes above and beyond what is currently available in the literature both in terms of survey paper, or the benchmarking of high-dimensional problems of interest.

Firstly, the manuscript fails to properly introduce the poli and poli-baseline libraries, and it fails to demonstrate how they can be used in practice to benchmark a discrete sequence optimization task. This hinders it's practicality to the wider BO community, and this is the main limitation of the paper.
Secondly, the benchmarking of HDBO tasks is not sufficient to warrant publication. The Benchmarking feels like an afterthought from the survey.

**Strengths:**

The paper is well written, and well structured throughout. The authors provide a taxonomy that allows the reader to navigate the complex field of HDBO methods. It also introduces new thematic areas on top of those introduced in recent literature, with key examples within the literature. However, I am not convinced that this adds much value.

**Additional Feedback:**

None

**Clarity:**

The paper is well written and structured. Although the survey takes up a considerable amount of the page count at the expense of the benchmarking.

**Correctness:**

I believe that the submission is correct, and the benchmarking was performed in a sound way – although I have not looked through the codebase.

**Documentation:**

There is a lack of detail about the software libraries introduced, the methods included, and the baseline tasks they contain.

**Limitations:**

Limitations were introduced briefly, such as a single experiment setting and limits on number of iterations, but no insights were provided as to how to go beyond this.

**Opportunities For Improvement:**

Whilst the Survey section of the paper is well motivated and structured, the Benchmarking section of the paper is underwhelming. This is the main opportunity to improve the paper.

There is a lack of information provided about the Protein Objective Library (poli) and the poli-baselines libraries. What baseline methods are included? What are the 35 tasks? What is the API, and how does this differ or augment existing libraries? How does a practitioner use the libraries?

Why were these particular methods benchmarked in Section 4.2? There are no insights raised about the benchmarking process. Whilst there is a brief section on page 7 that mentions GPyTorch, BoTorch, GPFlow and Trueste, I would expect a much more thorough comparison, since the benchmarking is presented as a key contribution of the paper. I would expect direct comparison to at least one additional software library.

Whilst I appreciate that optimization of the individual PMO task is not the contribution of the paper, there is no introduction of the problem being benchmarked. This section generally reads like an afterthought.

Finally, the benchmarking was performed on molecules with sequences of length 70. This does not go beyond the scale of the previous works, e.g. Santoni 2024, who benchmark problem sizes of 10, 20, 40 and 60. They also provide much more information regarding the compute costs, budgets and trade-offs inherent in HDBO. Why were no other Tasks, e.g. protein sequence design benchmarked? What are the limitations for going to larger sequences?

**Relation To Prior Work:**

The paper discusses many prior works.

**Summary And Contributions:**

This paper presents a unified view of the field of high-dimensional BO for biology and chemistry applications such as drug discovery. The authors present a detailed taxonomy of current methods. They also open source two benchmark packages (poli and poli-benchmark) that allow for consistent comparison between models and tasks. They finally benchmark 8 popular latent space BO methods using poli on a small molecule task.

---

> ### Author Rebuttal · Authors · 2024-08-16
>
> We thank the reviewer for the thorough feedback and constructive criticism, and we are glad to hear that our work reads well and is well-structured.
>
> In what follows, we would like to address the opportunities for improvement while quoting your review:
>
> > *Whilst the Survey section of the paper is well motivated and structured, the Benchmarking section of the paper is underwhelming. This is the main opportunity to improve the paper.*
>
> We thank the reviewer for underlining the structure and motivation of our survey, and we agree that our current benchmarking suite could be improved. With this goal in mind, we are currently running experiments on protein sequence optimization as we describe in the general rebuttal. We plan to elucidate the impact of sequence length and problem complexity à la Santoni et al. 2024, where the authors vary sequence lengths and provide information on wall times.
>
> > *There is a lack of information provided about the Protein Objective Library (poli) and the poli-baselines libraries. What baseline methods are included? What are the 35 tasks? What is the API, and how does this differ or augment existing libraries? How does a practitioner use the libraries?*
>
> This information was detailed in the documentation for poli and poli-baselines, but we acknowledge that this was not sufficiently clearly explained, and that these two libraries should be discussed in greater depth. In the new version, we plan to use half of the additional content to introduce their black boxes, methods, interface, and how they stand out from other software, including [a link to the documentation](https://machinelearninglifescience.github.io/poli-docs/) for specifics. An initial draft of said content can be found in the main rebuttal.
>
> > *Why were these particular methods benchmarked in Section 4.2?*
>
> The paper indeed does not argue for the selection of methods, and we thank the reviewer for pointing this out. We selected representative methods from the taxonomy we constructed through our survey, with the specific goal of determining whether methods that work directly on structured space fared against methods that rely on unsupervised information. We will update L257-261 accordingly and plan to gradually grow the collection of methods tested through our leaderboards, hoping that a community forms.
>
> > *There are no insights raised about the benchmarking process. Whilst there is a brief section on page 7 that mentions GPyTorch, BoTorch, GPFlow and Trueste, I would expect a much more thorough comparison, since the benchmarking is presented as a key contribution of the paper. I would expect direct comparison to at least one additional software library.*
>
> We thank the reviewer for their constructive criticism. We are addressing this comment in the following ways:
>
> - We now describe the software used by the different methods by including a table in the appendix (e.g. `vanilla_bo` using `Ax`, `botorch` and `gpytorch`, specifying the versions).
> - We include `pycma` and CMA-ES as another baseline for the latent space optimization in `poli-baselines`.
> - In future experiments added to our leaderboard, we will highlight compute costs, runtimes, and descriptions of the hardware. We are currently experiments on protein sequence design in as standardized an environment as possible (i.e. submitting identical jobs to a high-performance cluster).
>
> > *Whilst I appreciate that optimization of the individual PMO task is not the contribution of the paper, there is no introduction of the problem being benchmarked. This section generally reads like an afterthought.*
>
> We agree that the PMO task (and more generally small-molecule optimization) was introduced without context. We now include a description of the task in the appendix, highlighting how small molecules are represented as discrete SELFIES sequences.
>
> > *Finally, the benchmarking was performed on molecules with sequences of length 70\. This does not go beyond the scale of the previous works, e.g. Santoni 2024, who benchmark problem sizes of 10, 20, 40 and 60\. They also provide much more information regarding the compute costs, budgets and trade-offs inherent in HDBO. Why were no other Tasks, e.g. protein sequence design benchmarked? What are the limitations for going to larger sequences?*
>
> As we stated above, we are currently running experiments on protein sequence optimization exploring the impact of sequence length and problem complexity. Our work differs from Santoni et al 2024 in the sense that *we provide benchmarking tools* for discrete sequence optimization. Their work relies on an existing framework (i.e. COCO); in our work, we contribute a framework for the discrete setting that is easily extendable and which solves technical problems that arise when benchmarking several (Bayesian) optimization algorithms.

---

### Official Review · Reviewer_dDpj · 2024-07-21

**Rating:** 7
**Confidence:** 4
**Correctness:** Please see the Review textbox.
**Clarity:** Please see the Review textbox.

**Review:**

- This paper is generally well-written.
- Because this work heavily focuses on a survey, I don't know it has a good fit to the datasets and benchmarks track of NeurIPS. Now, I am on a positive side, but if other reviewers or area chairs also think that it doesn't fit to this track, I can agree with that.
- Figure 2: A y-axis label should be added. If possible, the addition of an x-axis label also helps to understand the diagram.
- The description of the benchmarks in terms of software development should be included more in the main article.
- I think that the authors missed this citation:

Kim, Jungtaek, Seungjin Choi, and Minsu Cho. "Combinatorial Bayesian optimization with random mapping functions to convex polytopes." Uncertainty in Artificial Intelligence. PMLR, 2022.

- Figure 3: It also needs axis labels.
- The societal impact written in the paper is too unclear. I think the authors can deliberately discuss it.

**Strengths:**

Please see the Review textbox.

**Additional Feedback:**

Please see the Review textbox.

**Documentation:**

Please see the Review textbox.

**Ethics:**

There is no specific ethical concern.

**Limitations:**

There are no particular societal limitations of this work.

**Opportunities For Improvement:**

Please see the Review textbox.

**Relation To Prior Work:**

Please see the Review textbox.

**Summary And Contributions:**

- This work provides a survey and benchmark of high-dimensional discrete Bayesian optimization.
- In particular, it covers the survey of high-dimensional Bayesian optimization for discrete sequences from diverse perspectives such as a taxonomy and the number of dimensions.
- Then, it proposes a benchmark for such a Bayesian optimization setting.

---

> ### Author Rebuttal · Authors · 2024-08-16
>
> We thank the reviewer for the constructive feedback. In particular, thanks for pointing out the reference we missed (which we have already added to Fig. 1 and Table 3, see [our project website’s about page](https://machinelearninglifescience.github.io/hdbo\_benchmark/docs/hdbo/introduction/) and [this link](https://www.miguelgondu.com/assets/hdbo\_timeline.pdf) for interactive updated versions of Fig. 1). We will also address the mishaps with Fig. 2 and 3, adding axis labels.
>
> Let us highlight the relevant opportunities for improvement from the review, and discuss our plans for addressing them.
>
> > *Because this work heavily focuses on a survey, I don't know it has a good fit to the datasets and benchmarks track of NeurIPS. Now, I am on a positive side, but if other reviewers or area chairs also think that it doesn't fit to this track, I can agree with that.*
>
> We acknowledge the concern about scope regarding our paper. As we discuss in the general rebuttal, we argue that our paper is in scope for the datasets and benchmarks track since we provide benchmarking tools, and apply them to the field of high-dimensional Bayesian optimization. Our paper tries to standardize the field, which is currently struggling to provide thorough and comparable experiments. From our perspective, the survey is an essential part of this goal as it provides the overview needed to ensure better experimental coverage in future publications.
>
> > *The description of the benchmarks in terms of software development should be included more in the main article.*
>
> We agree with the reviewer that more content space should be allocated for discussing our software. Indeed, we will expand Sec. 4.1. to include a discussion on the software as we highlight in the general rebuttal.
>
> > *The societal impact written in the paper is too unclear. I think the authors can deliberately discuss it.*
>
> We thank you for highlighting this. We have expanded the discussion on societal impact by diving deeper into the potential of drug discovery and its positive impact on society, as well as mentioning specific perils of dual-use (e.g. bioweaponry). Here is a draft of the revised version of the societal impact (lines 301-302):
>
> *Our benchmark and survey cover algorithms for discrete optimization, which has found application in areas of great positive societal impact (e.g. drug discovery and protein engineering); such tools could also be used for negative impacts, such as the development of bioweaponry.*

---

> > ### Comment · Reviewer_dDpj · 2024-08-20
> >
> > The authors' response has resolved my concerns.
> >
> > I maintain my score as accept, and I will be happy if this work is presented at NeurIPS.

---

### Official Review · Reviewer_eK8S · 2024-07-22
**Good survey of existing literature, but experiment choice could be improved**

**Rating:** 8
**Confidence:** 4
**Correctness:** yes, as far as I can tell
**Clarity:** Very clear

**Review:**

By surveying high-dimensional BO methods, this paper makes a significant contribution to the field by organizing and categorizing a bunch of related methods in a sub-field which lacks unifying concepts or directions (in my opinion). Figure 1 was especially informative. I think the discussion and classification of prior works was excellent, even though it was a bit short (which is understandable given the NeurIPS page limit).

The experiments were well-executed. The authors' code looks professional and well-maintained. The authors overcame several technical challenges from incompatible package versions to be able to benchmark everything consistently, and were able to test a very large number of methods. Their experiments in PMO also demonstrated very good attention to detail (for example they noticed inconsistencies between oracle outputs in different versions of the TDC package).

That being said, the focus of the paper is a bit confusing. What exactly is meant by "high-dimensional" when referring to sequences? Is it (sequence length) $\times$ (alphabet size)? Or do they just mean "long sequences"? It seems like they are conflating methods for several different tasks (e.g. high-dimensional BO in $\mathbb{R}^n$, BO over sequences, BO over graphs).

Also, the choice of experiments seem a little bit odd to me. Dockstring and PMO are all _graph_ design tasks, not sequence design tasks. Although one can encode a graph as a sequence (e.g. using SELFIES), it is a less fundamental descriptor. I thought that protein design would be a much more suitable task.

Overall, despite some issues with the experiments and the scope, I am happy to recommend acceptance. I think it would be inappropriate to reject a paper with this degree of thoroughness and engagement with prior literature.

**Strengths:**

- Focus is an important topic
- Great survey of large number of methods
- Code integrates a very large number of methods and is polished
- Paper is well-written
- Point out inconsistencies in prior works' evaluations (Figure 3, this was really nice)

**Additional Feedback:**

Overall this paper was a pleasure to read. However, I almost think that a short 9 page conference paper could be the wrong venue for this kind of review, because you don't have a lot of space to compare/contrast the methods. Given that you have already done so much work reviewing past literature, maybe it would make sense to write a more extended discussion with your insights in the appendix? This could be immensely valuable to people in the field.

**Documentation:**

Yes, well-documented.

**Ethics:**

No concerns.

**Limitations:**

yes, limitations discussed

**Opportunities For Improvement:**

- Be clearer on what you are benchmarking: is it sequence design, graph design, all sorts of structured design? Perhaps highlight the difference between "high dimensional BO" and "BO over structured spaces"
- PMO results are dominated by the logP and SA tasks, which are generally the simplest tasks (both can be optimized by producing large molecules). I would remove these: there is a reason they are not included in the main PMO benchmark.
- I would change the main experiment to be a sequence design task
- Experiments test only a fairly short horizon

**Relation To Prior Work:**

As a survey, this is very clear. Figure 1 is an excellent summary of prior work. I encourage the authors to release the TikZ code for this, if they indeed used TikZ.

**Summary And Contributions:**

This paper examines methods for Bayesian optimization over discrete sequences. It reviews and classifies a large amount of BO methods, then benchmarks them in a consistent way on a modified version of the PMO benchmark.

---

> ### Author Rebuttal · Authors · 2024-08-16
>
> We thank you for your input and constructive feedback. We are glad to hear that you believe our work engages thoroughly with prior literature, and we plan to improve our work based on your feedback.
>
> In particular, let us dive deeper into our plans while quoting the relevant parts of your review:
>
> > *What exactly is meant by "high-dimensional" when referring to sequences? Is it (sequence length)x(alphabet size)? Or do they just mean "long sequences"? It seems like they are conflating methods for several different tasks (e.g. high-dimensional BO in*
> *, BO over sequences, BO over graphs).*
>
> > *Be clearer on what you are benchmarking: is it sequence design, graph design, all sorts of structured design? Perhaps highlight the difference between "high dimensional BO" and "BO over structured spaces"*
>
> We apologize for the lack of clarity on this front, and thank the reviewer for highlighting this issue. The focus of our work has been optimizing functions of the sort $f\\colon \\mathcal{X}\\to\\mathbb{R}$, where $\\mathcal{X}$ is a space of sentences over an alphabet, i.e. *discrete sequence optimization*. We treat the PMO tasks as a discrete sequence optimization task through the choice of representing small molecules as SELFIES strings.
>
> Our survey finds that there are several potential ways in which discrete sequence optimization can be performed using Bayesian Optimization. Some methods work directly in the *structured space* (i.e. manipulating the discrete symbols either through continuous relaxations or kernel choices). Others, however, rely on learning latent representations and optimizing therein, casting a discrete problem into a continuous one. These continuous spaces open the doors for all other high-dimensional Bayesian optimization methods.
>
> By high-dimensional, we mean both (sequence length)x(alphabet size) as well as the size of the latent representations used for optimization.
>
> To remediate this misunderstanding we plan to add an extra figure to complement Fig. 1\. In it, we will describe the different ways in which a function $f\\colon\\mathcal{X}\\to\\mathbb{R}$ may be optimized using high-dimensional BO: (I) directly in discrete space as in the *structured spaces* family, (ii) using any of the high-dimensional methods over some continuous representation of the problem (e.g. one-hot encodings), and in particular (iii) through latent embeddings learned from unsupervised/supervised data.
>
> > *PMO results are dominated by the logP and SA tasks, which are generally the simplest tasks (both can be optimized by producing large molecules). I would remove these: there is a reason they are not included in the main PMO benchmark.*
>
> We agree that tasks like LogP and SA are easily exploitable and simple. We would still like to include them for completeness. It is important to provide this context, and we will modify Sec. 4.2. by adding a sentence about the exploitable nature of these black boxes.
>
> > *I would change the main experiment to be a sequence design task*.
>
> Thank you for your feedback. As we describe in the general rebuttal, we are currently experimenting with protein sequence optimization. The ongoing results of these experiments will become available in the project website, and we will point towards these changes as soon as they come.
>
> > *Experiments test only a fairly short horizon.*
>
> Our focus on short horizons comes from conversations with practitioners. When performing drug design or protein engineering, the cost of each experiment is so prohibitive that optimization loops are constrained to have budgets in the tens or low hundreds.
>
> That being said, we will accommodate for longer horizons in our Ehrlich function optimization experiment, given its closed form nature.
>
> > *As a survey, this is very clear. Figure 1 is an excellent summary of prior work. I encourage the authors to release the TikZ code for this, if they indeed used TikZ.*
>
> We are glad to hear you found Fig. 1 to be an excellent summary. We created Fig. 1 using the design software Figma after aggregating the survey. An SVG version of this figure can be found in our project’s website, which could be used for future updates.
>
> We believe our aggregated, interactive table would be as much a useful resource as Fig. 1 and Table 3, and we plan to release it open-source by building a page like [*What’s the score?*](https://scorebasedgenerativemodeling.github.io/) (a review of recent papers on score-based generative modeling) on our project’s website. [We have created a GitHub issue](https://github.com/MachineLearningLifeScience/hdbo\_benchmark/issues/12), and plan to work on it in future updates.
>
> > *Overall this paper was a pleasure to read. However, I almost think that a short 9 page conference paper could be the wrong venue for this kind of review, because you don't have a lot of space to compare/contrast the methods. Given that you have already done so much work reviewing past literature, maybe it would make sense to write a more extended discussion with your insights in the appendix? This could be immensely valuable to people in the field.*
>
> As we discuss in the main rebuttal, we believe this is an important point for discussion. We argue that our paper is within scope, and our ambition is for our project to become a hub of high-dimensional Bayesian optimization of discrete sequences. To this effect, we plan to blog continuously about the taxonomy and the methods therein.

---

### Official Review · Reviewer_wLu8 · 2024-07-24
**Impactful Benchmark; Questions on the Advantages of Poli Relative to Directly Evaluating on the Tasks Comprising it**

**Rating:** 7
**Confidence:** 5

**Review:**

&nbsp;

The authors introduce a comprehensive benchmark for one of the most important active areas of research in Bayesian optimization. The paper offers a comprehensive summary of the subfield and is well-written and organized. Overall, I am leaning towards acceptance but I would like to see more details in the paper on the exact composition of the benchmark as well as an elaboration of what aspects facilitate comparison between algorithms beyond the discussion in Section A.4 of the appendix. If the issues below can be addressed I will be inclined to increase my score.

&nbsp;

**Strengths:**

&nbsp;

The benchmark tackles an important problem, is comprehensive, and facilitates comparison between HDBO algorithms written in different programming frameworks.

&nbsp;

**Additional Feedback:**

&nbsp;

All relevant feedback has been provided above.

&nbsp;

**Clarity:**

&nbsp;

1. There are some missing capitalizations in the references e.g. "Bayesian".

2. Line 32, typo, "A benchmark of".

3. On line 163, in addition to pushing promising points together, metric learning encourages less promising points to be far from promising points, hence smoothening the latent function on the latent space, making it more amenable to GP fitting.

&nbsp;

**Correctness:**

&nbsp;

I see no technical problems with the current work although I did not attempt to reproduce the results of the experiments by running the authors' code.

&nbsp;

**Documentation:**

&nbsp;

The codebase is well-documented and, most-importantly, contains instructions for implementing a custom solver.

&nbsp;

**Ethics:**

&nbsp;

No ethical concerns.

&nbsp;

**Limitations:**

&nbsp;

The principle limitations as discussed above are the lack of emphasis in the paper on why poli and poli-baselines provide advantages relative to directly evaluating on the constituent benchmarks.

&nbsp;

**Opportunities For Improvement:**

&nbsp;

__**MAJOR POINTS**__

&nbsp;

The main weakness I see in the current work is that it does not appear to make a strong enough case for what advantages poli and polo-baselines afford over the PMO benchmark of Gao et al. For a HDBO practitioner, what is the salient difference between evaluating their algorithm on the PMO benchmark directly compared to the authors' benchmark? The current benchmark does not introduce new tasks for HDBO but rather aggregates existing benchmarks. While in principle, new benchmarks are not required, it would be great if the authors could edit the paper so as to more strongly emphasize the raison d'être for poli and poli-baselines and the advantages it provides for the community of practitioners relative to evaluating on the constituent benchmarks.

&nbsp;

__**MINOR POINTS**__

&nbsp;

1. On line 19, it may be worth adding references for the cited applications of Bayesian optimization e.g. drug discovery [1, 2, 3], protein design [4], machine learning hyper parameter tuning [5, 6] and train scheduling.

2. For Figure 1 and Table 3, some missing references that may warrant inclusion are [7-12].

3. It may be worth adding a genetic algorithm baseline to the results as per [19].

4. Is it possible to expand the remit of the benchmark to include antibody design [20]?


&nbsp;

&nbsp;

__**REFERENCES**__

&nbsp;

[1] Gómez-Bombarelli, R., Wei, J.N., Duvenaud, D., Hernández-Lobato, J.M., Sánchez-Lengeling, B., Sheberla, D., Aguilera-Iparraguirre, J., Hirzel, T.D., Adams, R.P. and Aspuru-Guzik, A., 2018. Automatic chemical design using a data-driven continuous representation of molecules. ACS Central Science, 4(2), pp.268-276.

[2] Griffiths, R.R. and Hernández-Lobato, J.M., 2020. Constrained Bayesian optimization for automatic chemical design using variational autoencoders. Chemical science, 11(2), pp.577-586.

[3] Pyzer-Knapp, E.O., 2018. Bayesian optimization for accelerated drug discovery. IBM Journal of Research and Development, 62(6), pp.2-1.

[4] Stanton, S., Maddox, W., Gruver, N., Maffettone, P., Delaney, E., Greenside, P. and Wilson, A.G., 2022, June. Accelerating Bayesian optimization for biological sequence design with denoising autoencoders. In International Conference on Machine Learning (pp. 20459-20478). PMLR.

[5] Snoek, J., Larochelle, H. and Adams, R.P., 2012. Practical Bayesian optimization of machine learning algorithms. Advances in neural information processing systems, 25.

[6] Turner, R., Eriksson, D., McCourt, M., Kiili, J., Laaksonen, E., Xu, Z. and Guyon, I., 2021, August. Bayesian optimization is superior to random search for machine learning hyperparameter tuning: Analysis of the black-box optimization challenge 2020. In NeurIPS 2020 Competition and Demonstration Track (pp. 3-26). PMLR.

[7] Eissman, S., Levy, D., Shu, R., Bartzsch, S. and Ermon, S., 2018, January. Bayesian optimization and attribute adjustment. In Proc. 34th Conference on Uncertainty in Artificial Intelligence.

[8] Lee, S., Chu, J., Kim, S., Ko, J. and Kim, H.J., 2024. Advancing Bayesian optimization via learning correlated latent space. Advances in Neural Information Processing Systems, 36.

[9] Verma, E., Chakraborty, S. and Griffiths, R.R., 2022. High-Dimensional Bayesian optimization with invariance. In ICML Workshop on Adaptive Experimental Design and Active Learning.

[10] Notin, P., Hernández-Lobato, J.M. and Gal, Y., 2021. Improving black-box optimization in VAE latent space using decoder uncertainty. Advances in Neural Information Processing Systems, 34, pp.802-814.

[11] Griffiths, R.R., Klarner, L., Moss, H., Ravuri, A., Truong, S., Du, Y., Stanton, S., Tom, G., Rankovic, B., Jamasb, A. and Deshwal, A., 2024. GAUCHE: a library for Gaussian processes in chemistry. Advances in Neural Information Processing Systems, 36.

[12] Kristiadi, A., Strieth-Kalthoff, F., Skreta, M., Poupart, P., Aspuru-Guzik, A. and Pleiss, G., A Sober Look at LLMs for Material Discovery: Are They Actually Good for Bayesian Optimization Over Molecules?. In Forty-first International Conference on Machine Learning.

[13] VR Saltenis (1971). One Method of Multiextremum Optimization. Avtomatika i Vychislitel’naya Tekhnika (Automatic Control and Computer Sciences) 5(3):33–38.

[14] Garnett, R., 2023. Bayesian optimization. Cambridge University Press.

[15] Kusner, M.J., Paige, B. and Hernández-Lobato, J.M., 2017, July. Grammar variational autoencoder. In International conference on machine learning (pp. 1945-1954). PMLR.

[16] Jin, W., Barzilay, R., & Jaakkola, T. (2018, July). Junction tree variational autoencoder for molecular graph generation. In International conference on machine learning (pp. 2323-2332). PMLR.

[17] Siivola, E., Paleyes, A., González, J., & Vehtari, A. (2021). Good practices for Bayesian optimization of high dimensional structured spaces. Applied AI Letters, 2(2), e24.

[18] Tripp, A. and Hernández-Lobato, J.M., 2024. Diagnosing and fixing common problems in Bayesian optimization for molecule design. arXiv preprint arXiv:2406.07709.

[19] Tripp, A. and Hernández-Lobato, J.M., 2023. Genetic algorithms are strong baselines for molecule generation. arXiv preprint arXiv:2310.09267.

[20] Khan, A., Cowen-Rivers, A.I., Grosnit, A., Robert, P.A., Greiff, V., Smorodina, E., Rawat, P., Akbar, R., Dreczkowski, K., Tutunov, R. and Bou-Ammar, D., 2023. Toward real-world automated antibody design with combinatorial Bayesian optimization. Cell Reports Methods, 3(1).

&nbsp;

**Relation To Prior Work:**

&nbsp;

1. When citing EI, it may be worth referencing the originating work [13] as discussed by Garnett in [14].

2. When discussing why high-dimensional BO is difficult, it may be worth discussing the additional factor of the GP kernel that was the focus of [11] where the authors showed that binary vector representations of molecules up to 4000 dimensions could be efficiently optimized in a BO setting with the Tanimoto kernel in place of continuous kernels such as the Matern.

3. In Section 3.5 it may be worth discussing [2, 15, 16] which closely followed [1] and considered Bayesian optimization over the latent space of a VAE as well as the follow-up works [7, 10, 17]. In terms of more recent work, it would be worth mentioning [12] which applies similar ideas over LLM embeddings.

4. When discussing BO over structured spaces, it would be worth mentioning the GAUCHE library [11] which uses kernels over binary vector and count vector representations of molecules to perform high-dimensional Bayesian optimization.

5. The recent work of [18] may also be worth mentioning in the "other" paragraph in Section 3.7.

&nbsp;

**Summary And Contributions:**

&nbsp;

The authors introduce a benchmark suite for high-dimensional BO (HDBO) methods formed predominantly of the Practical Molecular Optimization (PMO) benchmark of Gao et al. 2022 but also containing tasks from protein sequence design. The main purported advantage of the benchmark is that it provides a consistent interface to evaluate HDBO methods that is agnostic to the programming framework used.

&nbsp;

---

> ### Author Rebuttal · Authors · 2024-08-16
>
> We thank the reviewer for their feedback on our work. The references you point out are highly relevant for Fig. 1 and Table 3, and we are grateful for them. We have already added them (an online & interactive version of the updated Fig. 1 can be found on [the about page of our project’s website](https://machinelearninglifescience.github.io/hdbo\_benchmark/docs/hdbo/introduction/), as well as in the [PDF version here](https://www.miguelgondu.com/assets/hdbo\_timeline.pdf)).
>
> In particular, we have added the references in the following categories of our taxonomy:
>
> - Eissman et al. 2018 was already in Fig. 1 and Table 3 as *Attribute adj.*
> - Lee et al 2023 was already in Fig. 1 and Table 3 as *CoBo*.
> - Verma, Chakraborty & Griffiths 2022: Non-linear embeddings, with a link to Weighted retraining.
> - Notin, Hernández-Lobato and Gal 2021: Non-linear embeddings, with a link to LSBO.
> - Griffiths et al 2023 (GAUCHE): Structured spaces.
> - Kristadi et al 2024: Non-linear embeddings.
> - Khan et al 2023: Structured spaces.
>
> We have also included a reference suggested by reviewer dDpj on combinatorial Bayesian optimization (Kim et al 2022).
>
> In terms of the opportunities for improvement you highlight, let us discuss our plans for addressing your review:
>
> > *The main weakness I see in the current work is that it does not appear to make a strong enough case for what advantages poli and polo-baselines afford over the PMO benchmark of Gao et al. For a HDBO practitioner, what is the salient difference between evaluating their algorithm on the PMO benchmark directly compared to the authors' benchmark? The current benchmark does not introduce new tasks for HDBO but rather aggregates existing benchmarks. While in principle, new benchmarks are not required, it would be great if the authors could edit the paper so as to more strongly emphasize the raison d'être for poli and poli-baselines and the advantages it provides for the community of practitioners relative to evaluating on the constituent benchmarks.*
>
> We agree that our framework should be emphasized further. As such, and as we discuss in the general rebuttal, we plan to expand Sec. 4.1. to show practitioners the benefits of using `poli` and `poli-baselines`, namely
>
> - Automatic logging of each call to the black box that accommodates the use of libraries like `wandb` and `mlflow` ([See the documentation for a simple example of our logging logic](https://machinelearninglifescience.github.io/poli-docs/using\_poli/the\_basics/defining\_an\_observer.html) using observers, and [examples in our repository](https://github.com/MachineLearningLifeScience/poli/tree/dev/examples/observers)).
> - Reproducible environments for both black boxes and solvers. The environments in which both black-boxes and solvers run are tested weekly using cronjobs in GitHub actions. This guarantees not only that environments build, but that the numerical values obtained from querying black boxes remain the same. We have updated the `readme.md` files of both [`poli`](https://github.com/MachineLearningLifeScience/poli) and [`poli-baselines`](https://github.com/MachineLearningLifeScience/poli-baselines) to explicitly show tests passing in the different environments.
> - Our framework is easily extendable for adding new problems and solvers.
>
> > *On line 19, it may be worth adding references for the cited applications of Bayesian optimization e.g. drug discovery \[1, 2, 3\], protein design \[4\], machine learning hyper parameter tuning \[5, 6\] and train scheduling. For Figure 1 and Table 3, some missing references that may warrant inclusion are \[7-12\]*.
>
> We thank the reviewer for pointing us to these references. They are a great addition to our body of reviewed work. We have updated Fig. 1 and Table 3 accordingly, and you can find the links above.
>
> > *It may be worth adding a genetic algorithm baseline to the results as per \[19\].*
>
> `poli-baselines` has a simple genetic algorithm based on `pymoo`’s implementation. We have included results using this simple baseline as part of this rebuttal (see [our project website](https://machinelearninglifescience.github.io/hdbo\_benchmark/benchmarks/)). Due to the limited time available, we have been forced to leave implementing Tripp and Hernández-Lobato’s `mol_ga` for future work.
>
> > *Is it possible to expand the remit of the benchmark to include antibody design \[20\]?*
>
> Thank you for pointing us to AntBO. The authors were thorough and released the oracle in [their open source implementation](https://github.com/huawei-noah/HEBO/tree/master/AntBO). Such a black box is an ideal candidate for `poli`, and we will include it in our future sprints. [We have created an issue in `poli`](https://github.com/MachineLearningLifeScience/poli/issues/244) detailing the specifics to track progress.
>
> Finally, we have addressed your comments in the *Relation to prior work* section. We thank you for thoroughly checking the collection of papers we revised and commenting on our presentation.

---

> > ### Comment · Reviewer_wLu8 · 2024-08-22
> > **Many Thanks to the Authors for their Rebuttal**
> >
> > &nbsp;
> >
> > Many thanks to the authors for their rebuttal. Although, I voted to accept the paper in my first review, the main criticism was the advantages of poli over its constituent tasks. In their rebuttal the authors have clarified that a) poli enables integration with wandb and mlflow b) provides benefits in enabling reproducibility of the environments and c) is easily extensible. Additionally, The authors have also provided a genetic algorithm baseline.
> >
> > Given the clarification on the advantages of poli relative to its constituent benchmarks, I am inclined to increase my score by one point and vote for the paper to be accepted.
> >
> > &nbsp;

---

### Author Rebuttal · Authors · 2024-08-16

We thank the reviewers for their attentive and constructive feedback on our work. We are encouraged by the positive comments on the readability and clarity of our paper, and we acknowledge the opportunities for improvement.

After reading the reviews, we find the following common threads for discussion and improvement:
1. Elaborating on the discussion of the software (`poli` and `poli-baselines`) in the main article, tutorializing it for users,
2. Improving and expanding the benchmarking suite, and
3. Arguing for how our work fits this venue in particular.

What follows are our plans for improving the paper and addressing these comments. We will also reply to specific comments by the reviewers in individual rebuttal replies.

## Including more information about `poli` and `poli-baselines`

As reviewers 1TmH, wLu8 and dDpj rightfully point out, the discussion of the software was originally relegated to the appendix. To make our software contributions clearer in the new version of the manuscript, we now have a dedicated section on these libraries. In particular, we plan to use half of the extra content page to expand Sec. 4.1. (which describes the software) by explaining its interface, arguing for its necessity, and listing an example usage.

We find that explaining *the interface itself* would ideally start a discussion on how benchmarks of high-dimensional Bayesian optimization should be performed, and we thank the reviewers for pointing us in this direction.

## On expanding the benchmarking suite

During the review process, `poli`, `poli-baselines,` and the `hdbo-benchmark` have been under constant development. For example, we have added Ehrlich functions (Stanton et al. 2024\) as a new black box for discrete sequence design ([link to documentation here](https://machinelearninglifescience.github.io/poli-docs/using\_poli/objective\_repository/ehrlich\_functions.html)). These are closed-form black boxes designed to serve as proxies for antibody design assays, mimicking epistatic effects.

We are currently experimenting with Ehrlich functions, FoldX and RaSP black boxes for protein sequence design, varying the problem in terms of sequence length as well as other parameters that govern the difficulty. This is a prime opportunity to study how the methods we test perform on problems of varying dimensionality and difficulty. We hypothesize that the *structured spaces* family will perform better than other methods working on one-hot space for low sequence lengths.

Due to the nature of the problems and solvers, gathering results for these tasks takes a significant amount of compute. We will present ongoing results on our [project’s website and leaderboards](https://machinelearninglifescience.github.io/hdbo\_benchmark/benchmarks/). We want to highlight that our contribution is providing benchmarking tools that are easily extensible. As a specific example of this, we have already included `CMA-ES` and `SAASBO` results for the 2 and 128-dimensional latent spaces of PMO, as well as a simple `GeneticAlgorithm` baseline proposed by reviewer wLu8. We will point the reviewers to any more additions and changes in these leaderboards regarding our ongoing experiments during the rebuttal phase.

(Stanton et al. 2024\) Closed-Form Test Functions for Biophysical Sequence Optimization Algorithms, by Samuel Stanton, Robert Alberstein, Nathan Frey, Andrew Watkins, and Kyunghyun Cho. [https://arxiv.org/abs/2407.00236](https://arxiv.org/abs/2407.00236)

## Scope \- Datasets & Benchmarks track of NeurIPS

Reviewers eK8S and dDpj discuss our fit to the datasets and benchmarks track. Although none of the reviewers ultimately recommend disqualifying our paper for this reason, we believe it is a relevant discussion.

During the preparation of the manuscript, we have had doubts about this issue ourselves. However, [the call](https://neurips.cc/Conferences/2024/CallForDatasetsBenchmarks) states that the track welcomes all work on data-centric machine learning research and open-source libraries and tools that enable or accelerate ML research. We believe that our survey, benchmark, and open-source tools have the potential to standardize and accelerate research on the optimization of discrete sequences and therefore belong to this category. Although providing a survey of earlier work is not normally part of the manuscripts in this track, we believe that it in our case is an important aspect of the benchmark, forming the basis for the selection of representative methods from categories in the taxonomy.

Our survey collects several methods, including links to their open-source implementations. We argue that this aggregation is useful for practitioners interested in the current state of the field, aiding reproducibility and accelerating future work. Our benchmark tests a subset of these methods on discrete sequence black-box, emphasizing the fact that their performance highly depends on the problem it is tested on, and on the available supervised (or unsupervised information). Finally, `poli` democratizes access to several discrete sequence black boxes, and `poli-baselines` provide implementations of state-of-the-art discrete sequence optimization methods. Both software libraries distinguish themselves from previous work by having reliable, reproducible environments that are tested weekly via GitHub actions, and a unifying interface for optimization.

To summarize, we provide a thorough analysis of the field of high-dimensional Bayesian optimization of discrete sequences, and reproducible benchmarking tools that are tested in said domain.

---

> ### Author Response · Authors · 2024-08-30
> **Update on rebuttal**
>
> We thank reviewers wLu8 and dDpj for engaging with our official rebuttal, and we look forward to the discussions that will take place between reviewers and area chairs.
>
> During the rest of this rebuttal phase, we have been improving on the benchmarking suite and on the resources we provide. In what follows we summarize these changes:
>
> ## On expanding the benchmarking suite
>
> Besides implementing the recently-proposed Ehrlich functions (Stanton et al 2024), we have also improved on our RaSP implementation. RaSP (Blaabjerg et al 2023\) is a deep learning surrogate of the thermal stability of proteins (as measured using Rosetta), which allows us to perform high-throughput stability prediction and optimization in reasonable running times.
>
> Originally, RaSP could only predict the effect of single mutations on a given wildtype. We have extended our implementation to also work on several mutations by assuming *additivity*. This means that RaSP can now be used as a drop-in replacement for FoldX, which takes a significant amount of time to simulate the effect of several mutations.
>
> To optimize RaSP sequences using continuous latent representations, we trained an autoencoder using ESM-2 embeddings (Lin et al. 2022). This autoencoder gives us a map from a low-dimensional latent space back to sequences, and can thus be used for latent space Bayesian Optimization. The results of benchmarking on this new sequence design task are ongoing and [can be seen on our project’s website](https://machinelearninglifescience.github.io/hdbo\_benchmark/benchmarks/). To the best of our knowledge, our benchmarking software is the first to democratize access to RaSP for multiple mutations (under the additivity assumption).
>
> To summarize all changes from the original paper:
>
> - We have included new results on `CMA-ES`, `SAASBO` and `GeneticAlgorithm` for the PMO benchmark.
> - We have implemented new black boxes inside `poli`, namely Ehrlich functions and an additive version of `RaSP`.
> - We have experimented with `RaSP` optimization for generating thermally stable proteins using a latent space learned from ESM-2 embeddings.
>
> (Blaabjerg et al 2023): Rapid Protein Stability Prediction Using Deep Learning Representations, eLife 12, 2023\.
> (Lin et al. 2022):  Evolutionary-scale prediction of atomic-level protein structure with a language model, Science, 2022\.
>
> ## Expanding the resources
>
> Reviewer eK8S invites us to expand on our taxonomy and provide more resources. With that in mind, [we have added a page to our website in which we maintain the list of papers in our survey](https://machinelearninglifescience.github.io/hdbo\_benchmark/related\_work/).

---

### Decision · Program_Chairs · 2024-09-26

**Decision:**

Accept (Poster)

**Comment:**

This paper offers a survey and benchmarking framework for high-dimensional Bayesian optimization (HDBO) of discrete sequences, focusing on applications in drug discovery and protein design. The paper provides a comprehensive taxonomy of HDBO methods and introduces open-source libraries (poli and poli-baselines) to facilitate reproducibility and standardization of benchmarks. While reviewers appreciated the clarity and well-structured survey, they found the benchmarking section lacking depth, particularly in task complexity and insights.

Strengths:
- Comprehensive survey and taxonomy of high-dimensional Bayesian optimization methods.
- Clear writing and structure, with useful visual representations of prior work.

Weaknesses:
- Benchmarking feels incomplete and lacks depth in terms of both task complexity and insights.
- The practical advantages of poli and poli-baselines relative to existing benchmarks need stronger emphasis.
- Limited discussion on societal impacts and the broader relevance of the work to the NeurIPS Datasets and Benchmarks track.

Overall assessment: While the paper has certain limitations in its benchmarking section, the strength of the survey and the open-source tools justifies its acceptance. Revisions should focus on improving the depth of the benchmarking section, clarifying the advantages of the proposed software libraries, and adding more challenging tasks to showcase the utility of the framework more effectively.